# TRAINING GANS WITH OPTIMISM

**Constantinos Daskalakis**[*]
MIT, EECS
costis@mit.edu

**Andrew Ilyas**[*]
MIT, EECS
ailyas@mit.edu

**Vasilis Syrgkanis**[*]
Microsoft Research
vasy@microsoft.com

**Haoyang Zeng**[*]
MIT, EECS
haoyangz@mit.edu

## ABSTRACT

We address the issue of limit cycling behavior in training Generative Adversarial Networks and propose the use of Optimistic Mirror Decent (OMD) for training Wasserstein GANs. Recent theoretical results have shown that optimistic mirror decent (OMD) can enjoy faster regret rates in the context of zero-sum games. WGANs is exactly a context of solving a zero-sum game with simultaneous no-regret dynamics. Moreover, we show that optimistic mirror decent addresses the limit cycling problem in training WGANs. We formally show that in the case of bi-linear zero-sum games the last iterate of OMD dynamics converges to an equilibrium, in contrast to GD dynamics which are bound to cycle. We also portray the huge qualitative difference between GD and OMD dynamics with toy examples, even when GD is modified with many adaptations proposed in the recent literature, such as gradient penalty or momentum. We apply OMD WGAN training to a bioinformatics problem of generating DNA sequences. We observe that models trained with OMD achieve consistently smaller KL divergence with respect to the true underlying distribution, than models trained with GD variants. Finally, we introduce a new algorithm, Optimistic Adam, which is an optimistic variant of Adam. We apply it to WGAN training on CIFAR10 and observe improved performance in terms of inception score as compared to Adam.

## 1 INTRODUCTION

Generative Adversarial Networks (GANs) (Goodfellow et al., 2014) have proven a very successful approach for fitting generative models in complex structured spaces, such as distributions over images. GANs frame the question of fitting a generative model from a data set of samples from some distribution as a zero-sum game between a Generator (G) and a discriminator (D). The Generator is represented as a deep neural network which takes as input random noise and outputs a sample in the same space of the sampled data set, trying to approximate a sample from the underlying distribution of data. The discriminator, also modeled as a deep neural network is attempting to discriminate between a true sample and a sample generated by the generator. The hope is that at the equilibrium of this zero-sum game the generator will learn to generate samples in a manner that is indistinguishable from the true samples and hence has essentially learned the underlying data distribution.

Despite their success at generating visually appealing samples when applied to image generation tasks, GANs are very finicky to train. One particular problem, raised for instance in a recent survey as a major issue (Goodfellow, 2017) is the instability of the training process. Typically training of GANs is achieved by solving the zero-sum game via running simultaneously a variant of a Stochastic Gradient Descent algorithm for both players (potentially training the discriminator more frequently than the generator).

---

[1]Code for our models is available at https://github.com/vsyrgkanis/optimistic_GAN_training
[*]These authors contribute equally to this work.

The latter amounts essentially to solving the zero-sum game via running no-regret dynamics for each player. However, it is known from results in game theory, that no-regret dynamics in zero-sum games can very often lead to limit oscillatory behavior, rather than converge to an equilibrium. Even in convex-concave zero-sum games it is only the average of the weights of the two players that constitutes an equilibrium and not the last-iterate. In fact recent theoretical results of Mertikopoulos et al. (2017) show the strong result that no variant of GD that falls in the large class of Follow-the-Regularized-Leader (FTRL) algorithms can converge to an equilibrium in terms of the last-iterate and are bound to converge to limit cycles around the equilibrium.

Averaging the weights of neural nets is a prohibitive approach in particular because the zero-sum game that is defined by training one deep net against another is not a convex-concave zero-sum game. Thus it seems essential to identify training algorithms that make the last iterate of the training be very close to the equilibrium, rather than only the average.

**Contributions.** In this paper we propose training GANs, and in particular Wasserstein GANs Arjovsky et al. (2017), via a variant of gradient descent known as Optimistic Mirror Descent. Optimistic Mirror Descent (OMD) takes advantage of the fact that the opponent in a zero-sum game is also training via a similar algorithm and uses the predictability of the strategy of the opponent to achieve faster regret rates. It has been shown in the recent literature that Optimistic Mirror Descent and its generalization of Optimistic Follow-the-Regularized-Leader (OFTRL), achieve faster convergence rates than gradient descent in convex-concave zero-sum games (Rakhlin & Sridharan, 2013a;b) and even in general normal form games (Syrgkanis et al., 2015). Hence, even from the perspective of faster training, OMD should be preferred over GD due to its better worst-case guarantees and since it is a very small change over GD.

Moreover, we prove the surprising theoretical result that for a large class of zero-sum games (namely bi-linear games), OMD actually converges to an equilibrium in terms of the last iterate. Hence, we give strong theoretical evidence that OMD can help in achieving the long sought-after stability and last-iterate convergence required for GAN training. The latter theoretical result is of independent interest, since solving zero-sum games via no-regret dynamics has found applications in many areas of machine learning, such as boosting (Freund & Schapire, 1996). Avoiding limit cycles in such approaches could help improve the performance of the resulting solutions.

We complement our theoretical result with toy simulations that portray exactly the large qualitative difference between OMD as opposed to GD (and its many variants, including gradient penalty, momentum, adaptive step size etc.). We show that even in a simple distribution learning setting where the generator simply needs to learn the mean of a multi-variate distribution, GD leads to limit cycles, while OMD converges pointwise.

Moreover, we give a more complex application to the problem of learning to generate distributions of DNA sequences of the same cellular function. DNA sequences that carry out the same function in the genome, such as binding to a specific transcription factor, follow the same nucleotide distribution. Characterizing the DNA distribution of different cellular functions is essential for understanding the functional landscape of the human genome and predicting the clinical consequence of DNA mutations (Zeng et al., 2015; 2016; Zeng & Gifford, 2017). We perform a simulation study where we generate samples of DNA sequences from a known distribution. Subsequently we train a GAN to attempt to learn this underlying distribution. We show that OMD achieves consistently better performance than GD variants in terms of the Kullback-Leibler (KL) divergence between the distribution learned by the Generator and the true distribution.

Finally, we apply optimism to training GANs for images and introduce the *Optimistic Adam* algorithm. We show that it achieves better performance than Adam, in terms of inception score, when trained on CIFAR10.

## 2 PRELIMINARIES: WGANS AND OPTIMISTIC MIRROR DESCENT

We consider the problem of learning a generative model of a distribution of data points $Q \in \Delta(X)$. Our goal is given a set of samples from $D$, to learn an approximation to the distribution $Q$ in the form of a deep neural network $G_\theta(\cdot)$, with weight parameters $\theta$, that takes as input random noise

$z \in F$ (from some simple distribution $F$) and outputs a sample $G_\theta(z) \in X$. We will focus on addressing this problem via a Generative Adversarial Network (GAN) training strategy.

The GAN training strategy defines as a zero-sum game between a *generator* deep neural network $G_\theta(\cdot)$ and a *discriminator* neural network $D_w(\cdot)$. The generator takes as input random noise $z \sim F$, and outputs a sample $G_\theta(z) \in X$. A discriminator takes as input a sample $x$ (either drawn from the true distribution $Q$ or from the generator) and attempts to classify it as real or fake. The goal of the generator is to fool the discriminator.

In the original GAN training strategy Goodfellow et al. (2014), the discriminator of the zero sum game was formulated as a classifier, i.e. $D_w(x) \in [0, 1]$ with a multinomial logistic loss. The latter boils down to the following expected zero-sum game (ignoring sampling noise).

$$\inf_\theta \sup_w \mathbb{E}_{x \sim Q}\left[\log(D_w(x))\right] + \mathbb{E}_{z \sim F}\left[\log(1 - D_w(G_\theta(z)))\right] \tag{1}$$

If the discriminator is very powerful and learns to accurately classify all samples, then the problem of the generator amounts to minimizing the Jensen-Shannon divergence between the true distribution and the generators distribution. However, if the discriminator is very powerful, then in practice the latter leads to vainishing gradients for the generator and inability to train in a stable manner.

The latter problem lead to the formulation of Wasserstein GANs (WGANs) Arjovsky et al. (2017), where the discriminator rather than being treated as a classifier (equiv. approximating the JS divergence) is instead trying to approximate the Wasserstein$-1$ or *earth-mover* metric between the true distribution and the distribution of the generator. In this case, the function $D_w(x)$ is not constrained to being a probability in $[0, 1]$ but rather is an arbitrary 1-Lipschitz function of $x$. This reasoning leads to the following zero-sum game:

$$\inf_\theta \sup_w \mathbb{E}_{x \sim Q}\left[D_w(x)\right] - \mathbb{E}_{z \sim F}\left[D_w(G_\theta(z))\right] \tag{2}$$

If the function space of the discriminator covers all 1-Lipschitz functions of $x$, then the quantity $\sup_w \mathbb{E}_{x \sim D}\left[D_w(x)\right] - \mathbb{E}_{z \sim F}\left[D_w(G_\theta(z))\right]$ that the generator is trying to minimize corresponds to the earth-mover distance between the true distribution $Q$ and the distribution of the generator. Given the success of WGANs we will focus on WGANs in this paper.

## 2.1 Gradient Descent vs Optimistic Mirror Descent

The standard approach to training WGANs is to train simultaneously the parameters of both networks via stochastic gradient descent. We begin by presenting the most basic version of adversarial training via stochastic gradient descent and then comment on the multiple variants that have been proposed in the literature in the following section, where we compare their performance with our proposed algorithm for a simple example.

Let us start how training a GAN with gradient descent would look like in the absence of sampling error, i.e. if we had access to the true distribution $Q$. For simplicity of notation, let:

$$L(\theta, w) = \mathbb{E}_{x \sim Q}\left[D_w(x)\right] - \mathbb{E}_{z \sim F}\left[D_w(G_\theta(z))\right] \tag{3}$$

denote the loss in the expected zero-sum game of WGAN, as defined in Equation (2), i.e. $\inf_\theta \sup_w L(\theta, w)$. The classic WGAN approach is to solve this game by running gradient descent (GD) for each player, i.e. for $t \in \{1, \dots, T-1\}$: with $\nabla_{w,t} = \nabla_w L(\theta_t, w_t)$ and $\nabla_{\theta,t} = \nabla_\theta L(\theta_t, w_t)$

$$\begin{aligned} w_{t+1} &= w_t + \eta \cdot \nabla_{w,t} \\ \theta_{t+1} &= \theta_t - \eta \cdot \nabla_{\theta,t} \end{aligned} \tag{4}$$

If the loss function $L(\theta, w)$ was convex in $\theta$ and concave $w$, $\theta$ and $w$ lie in some bounded convex set and the step size $\eta$ is chosen of the order $\frac{1}{\sqrt{T}}$, then standard results in game theory and no-regret learning (see e.g. Freund & Schapire (1999)) imply that the pair $(\bar{\theta}, \bar{w})$ of average parameters, i.e. $\bar{w} = \frac{1}{T}\sum_{t=1}^{T} w_t$ and $\bar{\theta} = \frac{1}{T}\sum_{t=1}^{T} \theta_t$ is an $\epsilon$-equilibrium of the zero-sum game, for $\epsilon = O\left(\frac{1}{\sqrt{T}}\right)$. However, no guarantees are known beyond the convex-concave setting and, more importantly for the paper, even in convex-concave games, no guarantees are known for the last-iterate pair $(\theta_T, w_T)$.

Rakhlin and Sridharan (Rakhlin & Sridharan, 2013a) proposed an alternative algorithm for solving zero-sum games in a decentralized manner, namely *Optimistic Mirror Descent* (OMD), that achieves

faster convergence rates to equilibrium of $\epsilon = O\left(\frac{1}{T}\right)$ for the average of parameters. The algorithm essentially uses the last iterations gradient as a predictor for the next iteration's gradient. This follows from the intuition that if the opponent in the game is using a stable (or regularized) algorithm, then the gradients between the two iterations will not change much. Later Syrgkanis et al. (2015) showed that this intuition extends to show faster convergence of each individual player's regret in general normal form games.

Given these favorable properties of OMD when learning in games, we propose replacing GD with OMD when training WGANs. The update rule of a OMD is a small adaptation to GD. OMD is parameterized by a predictor of the next iteration's gradient which could either be simply last iteration's gradient or an average of a window of last gradient or a discounted average of past gradients. In the case where the predictor is simply the last iteration gradient, then the update rule for OMD boils down to the following simple form:

$$
\begin{aligned}
w_{t+1} &= w_t + 2\eta \cdot \nabla_{w,t} - \eta \cdot \nabla_{w,t-1} \\
\theta_{t+1} &= \theta_t - 2\eta \cdot \nabla_{\theta.t} + \eta \cdot \nabla_{\theta,t-1}
\end{aligned}
\tag{5}
$$

The simple modification in the GD update rule, is inherently different than any of the existing adaptations used in GAN training, such as Nesterov's momentum, or gradient penalty.

**General OMD and intuition.** The intuition behind OMD can be more easily understood when GD is viewed through the lens of the Follow-the-Regularized-Leader formulation. In particular, it is well known that GD is equivalent to the Follow-the-Regularized-Leader algorithm with an $\ell_2$ regularizer (see e.g. Shalev-Shwartz (2012)), i.e.:

$$
\begin{aligned}
w_{t+1} &= \arg\max_w \eta \sum_{\tau=1}^{t} \langle w, \nabla_{w,\tau} \rangle - \|w\|_2^2 \\
\theta_{t+1} &= \arg\min_\theta \eta \sum_{\tau=1}^{t} \langle \theta, \nabla_{\theta,\tau} \rangle + \|\theta\|_2^2
\end{aligned}
\tag{6}
$$

It is known that if the learner knew in advance the gradient at the next iteration, then by adding that to the above optimization would lead to constant regret that comes solely from the regularization term[1]. OMD essentially augments FTRL by adding a predictor $M_{t+1}$ of the next iterations gradient, i.e.:

$$
\begin{aligned}
w_{t+1} &= \arg\max_w \ \eta \left( \sum_{\tau=1}^{t} \langle w, \nabla_{w,\tau} \rangle + \langle w, M_{w,t+1} \rangle \right) - \|w\|_2^2 \\
\theta_{t+1} &= \arg\min_\theta \ \eta \left( \sum_{\tau=1}^{t} \langle \theta, \nabla_{\theta,\tau} \rangle + \langle \theta, M_{\theta,t+1} \rangle \right) + \|\theta\|_2^2
\end{aligned}
\tag{7}
$$

For an arbitrary set of predictors, the latter boils down to the following set of update rules:

$$
\begin{aligned}
w_{t+1} &= w_t + \eta \cdot (\nabla_{w,t} + M_{w,t+1} - M_{w,t}) \\
\theta_{t+1} &= \theta_t - \eta \cdot (\nabla_{\theta,t} + M_{\theta,t+1} - M_{\theta,t})
\end{aligned}
\tag{8}
$$

In the theoretical part of the paper we will focus on the case where the predictor is simply the last iteration gradient, leading to update rules in Equation (5). In the experimental section we will also explore performance of other alternatives for predictors.

## 2.2 Stochastic Gradient Descent and Stochastic Optimistic Mirror Descent

In practice we don't really have access to the true distribution $Q$ and hence we replace $Q$ with an empirical distribution $Q_n$ over samples $\{x_1, \ldots, x_n\}$ and $F_n$ of random noise samples $\{z_1, \ldots, z_n\}$, leading to empirical loss for the zero-sum game of:

$$
L_n(\theta, w) = \mathbb{E}_{x \sim Q_n} [D_w(x)] - \mathbb{E}_{z \sim F_n} [D_w(G_\theta(z))]
\tag{9}
$$

Even in this setting it might be impractical to compute the gradient of the expected loss with respect to $Q_n$ or $F_n$, e.g. $\mathbb{E}_{x \sim Q_n} [\nabla_w D_w(x)]$.

---

[1]The latter is a consequence of the be-the-leader lemma Kalai & Vempala (2005); Rigollet (2015)

However, GD and OMD still leads to small loss if we replace gradients with unbiased estimators of them. Hence, we can replace expectation with respect to $Q_n$ or $F_n$, by simply evaluating the gradients at a single sample or on a small batch of $B$ samples. Hence, we can replace the gradients at each iteration with the variants:

$$
\begin{aligned}
\hat{\nabla}_{w,t} &= \frac{1}{|B|} \sum_{i \in B} \left( \nabla_w D_{w_t}(x_i) - \nabla_w D_{w_t}(G_{\theta_t}(z_i)) \right) \\
\hat{\nabla}_{\theta,t} &= -\frac{1}{|B|} \sum_{i \in B} \nabla_\theta (D_{w_t}(G_{\theta_t}(z_i)))
\end{aligned}
\tag{10}
$$

Replacing $\nabla_{w,t}$ and $\nabla_{\theta,t}$ with the above estimates in Equation (4) and (5), leads to Stochastic Gradient Descent (SGD) and Stochastic Optimistic Mirror Decent (SOMD) correspondingly.

## 3  AN ILLUSTRATIVE EXAMPLE: LEARNING THE MEAN OF A DISTRIBUTION

We consider the following very simple WGAN example: The data are generated by a multivariate normal distribution, i.e. $Q \triangleq N(v, I)$ for some $v \in \mathbb{R}^d$. The goal is for the generator to learn the unknown parameter $v$. In Appendix C we also consider a more complex example where the generator is trying to learn a co-variance matrix.

We consider a WGAN, where the discriminator is a linear function and the generator is a simple additive displacement of the input noise $z$, which is drawn from $F \triangleq N(0, I)$, i.e:

$$
\begin{aligned}
D_w(x) &= \langle w, x \rangle \\
G_\theta(z) &= z + \theta
\end{aligned}
\tag{11}
$$

The goal of the generator is to figure out the true distribution, i.e. to converge to $\theta = v$. The WGAN loss then takes the simple form:

$$
L(\theta, w) = \mathbb{E}_{x \sim N(v,I)} \left[ \langle w, x \rangle \right] - \mathbb{E}_{z \sim N(0,I)} \left[ \langle w, z + \theta \rangle \right]
\tag{12}
$$

We first consider the case where we optimize the true expectations above rather than assuming that we only get samples of $x$ and samples of $z$. Due to linearity of expectation, the expected zero-sum game takes the form:

$$
\inf_\theta \sup_w \ \langle w, v - \theta \rangle
\tag{13}
$$

We see here that the unique equilibrium of the above game is for the generator to choose $\theta = v$ and for the discriminator to choose $w = 0$. For this simple zero sum game, we have $\nabla_{w,t} = v - \theta_t$ and $\nabla_{\theta,t} = -w_t$. Hence, the GD dynamics take the form:

$$
\begin{aligned}
w_{t+1} &= w_t + \eta(v - \theta_t) \\
\theta_{t+1} &= \theta_t + \eta w_t
\end{aligned}
\qquad \text{(GD Dynamics for Learning Means)}
$$

while the OMD dynamics take the form:

$$
\begin{aligned}
w_{t+1} &= w_t + 2\eta \cdot (v - \theta_t) - \eta \cdot (v - \theta_{t-1}) \\
\theta_{t+1} &= \theta_t + 2\eta \cdot w_t - \eta \cdot w_{t-1}
\end{aligned}
\qquad \text{(OMD Dynamics for Learning Means)}
$$

We simulated simultaneous training in this zero-sum game under the GD and under OMD dynamics and we find that GD dynamics always lead to a limit cycle irrespective of the step size or other modifications. In Figure 1 we present the behavior of the GD vs OMD dynamics in this game for $v = (3, 4)$. We see that even though GD dynamics leads to a limit cycle (whose average does indeed equal to the true vector), the OMD dynamics converge to $v$ in terms of the last iterate. In Figure 2 we see that the stability of OMD even carries over to the case of Stochastic Gradients, as long as the batch size is of decent size.

In the appendix we also portray the behavior of the GD dynamics even when we add gradient penalty (Gulrajani et al., 2017) to the game loss (instead of weight clipping), adding Nesterov momentum to the GD update rule (Nesterov, 1983) or when we train the discriminator multiple times in between a train iteration of the generator. We see that even though these modifications do improve the stability

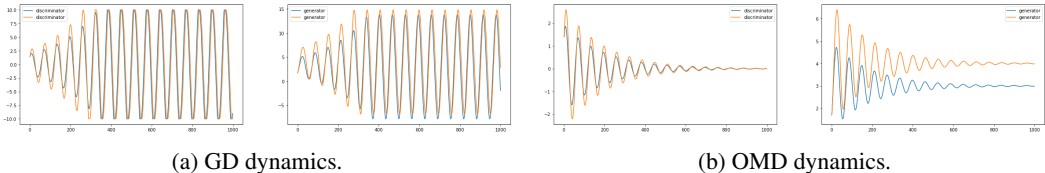

(a) GD dynamics.

(b) OMD dynamics.

Figure 1: Training GAN with GD converges to a limit cycle that oscilates around the equilibrium (we applied weight-clipping at 10 for the discriminator). On the contrary training with OMD converges to equilibrium in terms of last-iterate convergence.

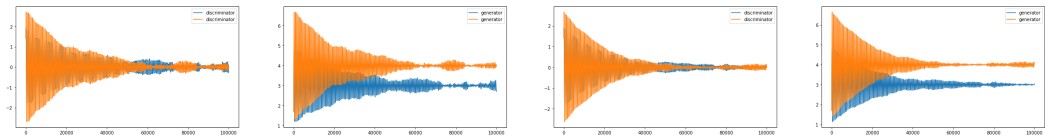

(a) Stochastic OMD dynamics with mini-batch of 50.    (b) Stochastic OMD dynamics with mini-batch of 200.

Figure 2: Robustness of last-iterate convergence of OMD to stochastic gradients.

of the GD dynamics, in the sense that they narrow the band of the limit cycle, they still lead to a non-vanishing limit cycle, unlike OMD.

In the next section, we will in fact prove formally that for a large class of zero-sum games including the one presented in this section, OMD dynamics converge to equilibrium in the sense of last-iterate convergence, as opposed to average-iterate convergence.

## 4    LAST-ITERATE CONVERGENCE OF OPTIMISTIC ADVERSARIAL TRAINING

In this section, we show that Optimistic Mirror Descent exhibits final-iterate, rather than only average-iterate convergence to min-max solutions for bilinear functions. More precisely, we consider the problem $\min_x \max_y x^T A y$, for some matrix $A$, where $x$ and $y$ are unconstrained. In Appendix D, we also show that our convergence result appropriately extends to the general case, where the bi-linear game also contains terms that are linear in the players' individual strategies, i.e. games of the form:

$$\inf_x \sup_y \left( x^T A y + b^T x + c^T y + d \right). \tag{14}$$

In the simpler $\min_x \max_y x^T A y$ problem, Optimistic Mirror Descent takes the following form, for all $t \geq 1$:

$$x_t = x_{t-1} - 2\eta A y_{t-1} + \eta A y_{t-2} \tag{15}$$

$$y_t = y_{t-1} + 2\eta A^T x_{t-1} - \eta A^T x_{t-2} \tag{16}$$

*Initialization:* For the above iteration to be meaningful we need to specify $x_0, x_{-1}, y_0, y_{-1}$. We choose any $x_0 \in \mathcal{R}(A)$, and $y_0 \in \mathcal{R}(A^T)$, and set $x_{-1} = 2x_0$ and $y_{-1} = 2y_0$, where $\mathcal{R}(\cdot)$ represents the column space of $A$. In particular, our initialization means that the first step taken by the dynamics gives $x_1 = x_0$ and $y_1 = y_0$.

We will analyze Optimistic Mirror Descent under the assumption $\lambda_\infty \leq 1$, where $\lambda_\infty = \max\{||A||, ||A^T||\}$ and $||\cdot||$ denotes spectral norm of matrices. We can always enforce that $\lambda_\infty \leq 1$ by appropriately scaling $A$. Scaling $A$ by some positive factor clearly does not change the min-max solutions $(x^*, y^*)$, only scales the optimal value $x^{*T} A y^*$ by the same factor.

We remark that the set of equilibrium solutions of this minimax problem are pairs $(x, y)$ such that $x$ is in the null space of $A^T$ and $y$ is in the null space of $A$. In this section we rigorously show that Optimistic Mirror Descent converges to the set of such min-max solutions. This is interesting in light of the fact that Gradient Descent actually diverges, even in the special case where $A$ is the identity matrix, as per the following proposition whose proof is provided in Appendix D.3.

**Proposition 1.** *Gradient descent applied to the problem* $\min_x \max_y x^T y$ *diverges starting from any initialization* $x_0, y_0$ *such that* $x_0, y_0 \neq 0$.

Next, we state our main result of this section, whose proof can be found in Appendix D, where we also state its appropriate generalization to the general case (14).

**Theorem 1** (Last Iterate Convergence of OMD). *Consider the dynamics of Eq.* (15) *and* (16) *and any initialization* $\frac{1}{2} x_{-1} = x_0 \in \mathcal{R}(A)$, *and* $\frac{1}{2} y_{-1} = y_0 \in \mathcal{R}(A^T)$. *Let also*

$$\gamma = \left\| \left( AA^T \right)^+ \right\|,$$

*where for a matrix* $X$ *we denote by* $X^+$ *its generalized inverse and by* $\|X\|$ *its spectral norm. Suppose that* $\|A\| \equiv \lambda_\infty \leq 1$ *and that* $\eta$ *is a small enough constant satisfying* $\eta < 1/(3\gamma^2)$. *Letting* $\Delta_t = \left\| A^T x_t \right\|_2^2 + \|A y_t\|_2^2$, *the OMD dynamics satisfy the following:*

$$\Delta_1 = \Delta_0 \geq \frac{1}{(1+\eta)^2} \Delta_2$$

$$\forall t \geq 3: \quad \Delta_t \leq \left( 1 - \frac{\eta^2}{\gamma^2} \right) \Delta_{t-1} + 16\eta^3 \Delta_0.$$

*In particular,* $\Delta_t \to O(\eta\gamma^2\Delta_0)$, *as* $t \to +\infty$, *and for large enough* $t$, *the last iterate of OMD is within* $O(\sqrt{\eta} \cdot \gamma \sqrt{\Delta_0})$ *distance from the space of equilibrium points of the game, where* $\sqrt{\Delta_0}$ *is the distance of the initial point* $(x_0, y_0)$ *from the equilibrium space, and where both distances are taken with respect to the norm* $\sqrt{x^T AA^T x + y^T A^T Ay}$.

## 5 EXPERIMENTAL RESULTS FOR GENERATING DNA SEQUENCES

We take our theoretical intuition to practice, applying OMD to the problem of generating DNA sequences from an observed distribution of sequences. DNA sequences that carry out the same function can be viewed as samples from some distribution. For many important cellular functions, this distribution can be well modeled by a position-weight matrix (PWM) that specifies the probability of different nucleotides occuring at each position (Stormo, 2000). Thus, training GANs from DNA sequences sampled from a PWM distribution serves as a practically motivated problem where we know the ground truth and can thus quantify the performance of different training methods in terms of the KL divergence between the trained generator distribution and the true distribution.

In our experiments, we generated 40,000 DNA sequences of six nucleotides according to a given position weight matrix. A random 10% of the sequences were held out as the validation set. Each sequence was then embedded into a $4 \times 6$ matrix by encoding each of the four nucleotides with an one-hot vector. On this dataset, we trained WGANs with different variants of OMD and SGD and evaluated their performance in terms of the KL divergence between the empirical distribution of the WGAN-generated samples and the true distribution described by the position weight matrix. Both the discriminator and generator of the WGAN used in this analysis were chosen to be convolutional neural networks (CNN), given the recent success of CNNs in modeling DNA-protein binding (Zeng et al., 2016; Alipanahi et al., 2015). The detailed structure of the chosen CNNs can be found in Appendix E.

To account for the impact of learning rate and training epochs, we explored two different ways of model selection when comparing different optimization strategies: (1) using the iteration and learning rate that yields the lowest discriminator loss on the held out test set. This is inspired by the observation in Arjovsky et al. (2017) that the discriminator loss negatively correlates with the quality of the generated samples. (2) using the model obtained after the last epoch of the training. To account for the stochastic nature of the initialization and optimizers, we trained 50 independent models for each learning rate and optimizer, and compared the optimizer strategies by the resulting distribution of KL divergences across 50 runs.

For GD, we used variants of Equation (4) to examine the effect of using momentum and an adaptive step size. Specifically, we considered momentum, Nesterov momentum and Adagrad. The specific form of all these modifications is given for reference in Appendix A.

For OMD we used the general predictor version of Equation (10) with a fixed step size and with the following variants of the next iteration predictor $M_{t+1}$: (v1) Last iteration gradient: $M_{t+1} = \nabla f_t$, (v2) Running average of past gradients: $M_{t+1} = \frac{1}{t}\sum_{i=1}^{t}\nabla f_i$, (v3) Hyperbolic discounted average of past gradients: $M_{t+1} = \lambda M_t + (1-\lambda)\nabla f_t, \lambda \in (0,1)$. We explored two training schemes: (1) training the discriminator 5 times for each generator training as suggest in Arjovsky et al. (2017). (2) training the discriminator once for each generator training. The latter is inline with the intuition behind the use of optimism: optimism hinges on the fact that the gradient at the next iteration is very predictable since it is coming from another regularized algorithm, and if we train the other algorithm multiple times, then the gradient is not that predictable and the benefits of optimism are lost.

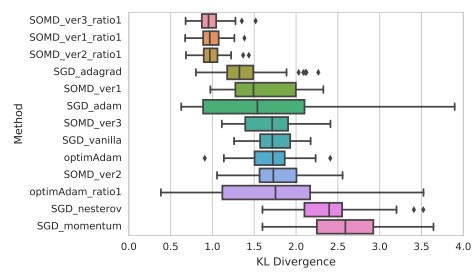

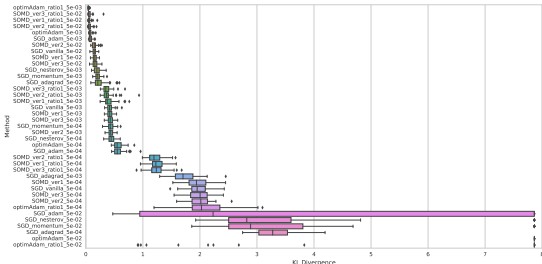

(a) WGAN with the lowest validation discriminator loss

(b) WGAN at the last epoch

Figure 3: KL divergence of WGAN trained with different optimization strategies. Methods are ordered by the median KL divergence. Methods in (a) are named by the category and version of the method. "ratio 1" denotes training the discriminator once for every generator training. Otherwise, we performed 5 iterations. For (b) where we don't combine the models trained with different learning rates, the learning rate is appended at the end of the method name. For momentum and Nesterov momentum, we used $\gamma = 0.9$. For Adagrad, we used the default $\epsilon = 1e^{-8}$.

For all afore-described algorithms, we experimented with their stochastic variants. Figure 3 shows the KL divergence between the WGAN-generated samples and the true distribution. When evaluated by the epoch and learning rate that yields the lowest discriminator loss on the validation set, WGAN trained with Stochastic OMD (SOMD) achieves lower KL divergence than the competing SGD variants. Evaluated by the last epoch, the best performance across different learning rates is achieved by optimistic Adam (see Section 6). We note that in both metrics, SOMD with 1:1 generator-discriminator training ratio yields better KL divergence than the alternative training scheme (1:5 ratio), which validates the intuition behind the use of optimism.

## 6 GENERATING IMAGES FROM CIFAR10 WITH OPTIMISTIC ADAM

In this section we applying optimistic WGAN training to generating images, after training on CIFAR10. Given the success of Adam on training image WGANs we will use an optimistic version of the Adam algorithm, rather than vanilla OMD. We denote the latter by *Optimistic Adam*. Optimistic Adam could be of independent interest even beyond training WGANs. We present Optimistic Adam for (G) but the analog is also used for training (D). We trained on CIFAR10 images with

---

**Algorithm 1** *Optimistic ADAM*, proposed algorithm for training WGANs on images.

---

Parameters: stepsize $\eta$, exponential decay rates for moment estimates $\beta_1, \beta_2 \in [0,1)$, stochastic loss as a function of weights $\ell_t(\theta)$, initial parameters $\theta_0$
**for** each iteration $t \in \{1, \dots, T\}$ **do**
    Compute stochastic gradient: $\nabla_{\theta,t} = \nabla_\theta \ell_t(\theta)$
    Update biased estimate of first moment: $m_t = \beta_1 m_{t-1} + (1-\beta_1) \cdot \nabla_{\theta,t}$
    Update biased estimate of second moment: $v_t = \beta_2 v_{t-1} + (1-\beta_2) \cdot \nabla_{\theta,t}^2$
    Compute bias corrected first moment: $\hat{m}_t = m_t/(1-\beta_1^t)$
    Compute bias corrected second moment: $\hat{v}_t = v_t/(1-\beta_2^t)$
    Perform *optimistic gradient step*: $\theta_t = \theta_{t-1} - 2\eta \cdot \frac{\hat{m}_t}{\sqrt{\hat{v}_t}+\epsilon} + \eta\frac{\hat{m}_{t-1}}{\sqrt{\hat{v}_{t-1}}+\epsilon}$

Return $\theta_T$

---

Optimistic Adam with the hyper-parameters matched to Gulrajani et al. (2017), and we observe that

it outperforms Adam in terms of inception score (see Figure 14), a standard metric of quality of WGANs (Gulrajani et al., 2017; Salimans et al., 2016). In particular we see that optimistic Adam achieves high numbers of inception scores after very few epochs of training. We observe that for Optimistic Adam, training the discriminator once after one iteration of the generator training, which matches the intuition behind the use of optimism, outperforms the 1:5 generator-discriminator training scheme. We see that vanilla Adam performs poorly when the discriminator is trained only once in between iterations of the generator training. Moreover, even if we use vanilla Adam and train 5 times (D) in between a training of (G), as proposed by Arjovsky et al. (2017), then performance is again worse than Optimistic Adam with a 1:1 ratio of training. The same learning rate 0.0001 and betas ($\beta_1 = 0.5, \beta_2 = 0.9$) as in Appendix B of Gulrajani et al. (2017) were used for all the methods compared. We also matched other hyper-parameters such as gradient penalty coefficient $\lambda$ and batch size. For a larger sample of images see Appendix G.

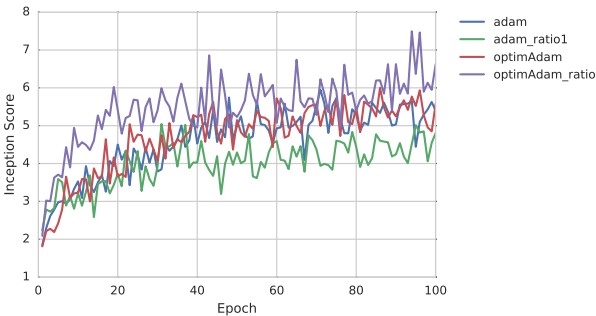

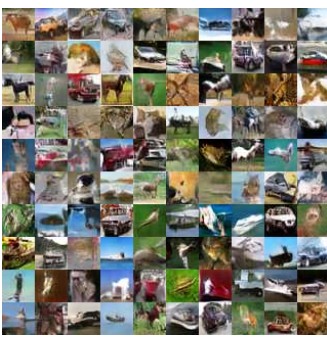

(a) Inception score on CIFAR10, when training with Adam and Optimistic Adam. "ratio1" means we performed 1 iteration of training of (D) in between 1 iteration of (G). Otherwise we performed 5 iterations. We further test (averaging over 35 trials) the two top-performing optimizers, Adam (ratio 5) and Optimistic Adam with ratio 1, in Appendix H.

(b) Sample of images from Generator of Epoch 94, which had the highest inception score.

Figure 4: Comparison of Adam and Optimistic Adam on CIFAR10.

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

## A  VARIANTS OF GD TRAINING

For ease of reference we briefly describe the exact form of update rules for several modifications of GD training that we have used in our experimental results.

Adagrad:

$$\eta_{w,t} = \frac{\eta}{\sqrt{\sum_{i=1}^{t} \nabla_{w,i}^2} + \epsilon} \qquad \eta_{\theta,t} = \frac{\eta}{\sqrt{\sum_{i=1}^{t} \nabla_{\theta,i}^2} + \epsilon}$$

$$w_{t+1} = w_t + \eta_{w,t} \cdot \nabla_{w,t} \qquad \theta_{t+1} = \theta_t - \eta_{\theta,t} \cdot \nabla_{\theta,t} \tag{17}$$

Momentum:

$$v_{w,t+1} = \gamma \cdot v_{w,t} + \eta \cdot \nabla_{w,t} \qquad v_{\theta,t+1} = \gamma \cdot v_{\theta,t} + \eta \cdot \nabla_{\theta,t}$$

$$w_{t+1} = w_t + v_{w,t+1} \qquad \theta_{t+1} = \theta_t - v_{\theta,t+1} \tag{18}$$

Nesterov momentum:

$$w_{\text{ahead}} = w_t + \gamma \cdot v_{w,t} \qquad \theta_{\text{ahead}} = \theta_t - \gamma \cdot v_{\theta,t}$$

$$v_{w,t+1} = \gamma \cdot v_{w,t} + \eta \cdot \nabla_w L(\theta_t, w_{\text{ahead}}) \qquad v_{\theta,t+1} = \gamma \cdot v_{\theta,t} + \eta \cdot \nabla_\theta L(\theta_{\text{ahead}}, w_t) \tag{19}$$

$$w_{t+1} = w_t + v_{w,t+1} \qquad \theta_{t+1} = \theta_t - v_{\theta,t+1}$$

## B  PERSISTENCE OF LIMIT CYCLES IN GD TRAINING

In Figure 5 we portray example Gradient Descent dynamics in the illustrative example described in Section 3 under multiple adaptations proposed in the literature. We observe that oscillations persist in all such modified GD dynamics, though alleviated by some. We briefly describe the modifications in detail first.

**Gradient penalty.**  The Wasserstein GAN is based on the idea that the discriminator is approximating all 1-Lipschitz functions of the data. Hence, when training the discriminator we need to make sure that the function $D_w(x)$ has a bounded gradient with respect to $x$. One approach to achieving this is weight-clipping, i.e. clipping the weights to lie in some interval. However, the latter might introduce extra instability during training. Gulrajani et al. (2017) introduce an alternative approach by adding a penalty to the loss function of the zero-sum game that is essentially the $\ell_2$ norm of the gradient of $D_w(x)$ with respect to $x$. In particular they propose the following regularized WGAN loss:

$$L_\lambda(\theta, w) = \mathbb{E}_{x \sim Q}\left[D_w(x)\right] - \mathbb{E}_{z \sim F}\left[D_w(G_\theta(z))\right] - \lambda \mathbb{E}_{\hat{x} \sim Q_\epsilon}\left[\left(\|\nabla_x D_w(\hat{x})\| - 1\right)^2\right]$$

where $Q_\epsilon$ is the distribution of the random vector $\epsilon x + (1 - \epsilon)G(z)$ when $x \sim Q$ and $z \sim F$. The expectations in the latter can also be replaced with sample estimates in stochastic variants of the training algorithms.

For our simple example, $\nabla_x D_w(x) = w$. Hence, we get the gradient penalty modified WGAN:

$$L_\lambda(\theta, w) = \langle w, v - \theta \rangle - \lambda \left(\|w\| - 1\right)^2 \tag{20}$$

Hence, the gradient of the modified loss function with respect to $\theta$ remains unchanged, but the gradient with respect to $w$ becomes:

$$\nabla_{wt} = v - \theta_t - 2\lambda w_t \frac{\|w_t\|_2 - 1}{\|w_t\|_2} \tag{21}$$

**Momentum.**  GD with momentum was defined in Equation (18). For the case of the simple illustrative example, these dynamics boil down to:

$$m_{w,t+1} = \gamma \cdot m_{w,t} + \eta \cdot (v - \theta_t) \qquad m_{\theta,t+1} = \gamma \cdot m_{\theta,t} - \eta \cdot w_t$$

$$w_{t+1} = w_t + m_{w,t+1} \qquad \theta_{t+1} = \theta_t - m_{\theta,t+1} \tag{22}$$

**Nesterov momentum.** GD with Nesterov's momentum was defined in Equation (19). For the illustrative example, we see that Nesterov's momentum is identical to momentum in the absence of gradient penalty. The reason being that the function is bi-linear. However, with a gradient penalty, Nesterov's momentum boils down to the following update rule.

$$\hat{w}_t = w_t + \gamma \cdot m_{w,t}$$

$$m_{w,t+1} = \gamma \cdot m_{w,t} + \eta \cdot (v - \theta_t) - 2\eta \cdot \lambda \hat{w}_t \frac{\|\hat{w}_t\|_2 - 1}{\|\hat{w}_t\|_2} \qquad m_{\theta,t+1} = \gamma \cdot m_{w,t} - \eta \cdot w_t \quad (23)$$

$$w_{t+1} = w_t + m_{w,t+1} \qquad\qquad\qquad \theta_{t+1} = \theta_t - m_{\theta,t+1}$$

**Asymmetric training.** Another approach to reducing cycling is to train the discriminator more frequently than the generator. Observe that if we could exactly solve the supremum problem of the discriminator after every iteration of the generator, then the generator would be simply solving a convex minimization problem and GD should converge point-wise. The latter approach could lead to slow convergence given the finiteness of samples in the case of stochastic training. Hence, we cannot really afford completely solving the discriminators problem. However, training the discriminator for multiple iterations, brings the problem faced by the generator closer to convex minimization rather than solving an equilibrium problem. Hence, asymmetric training could help with cycling. We observe below that asymmetric training is the most effective modification in reducing the range of the cycles and hence making the last-iterate be close to the equilibrium. However, it does not really eliminate the cycles, rather it simply makes their range smaller.

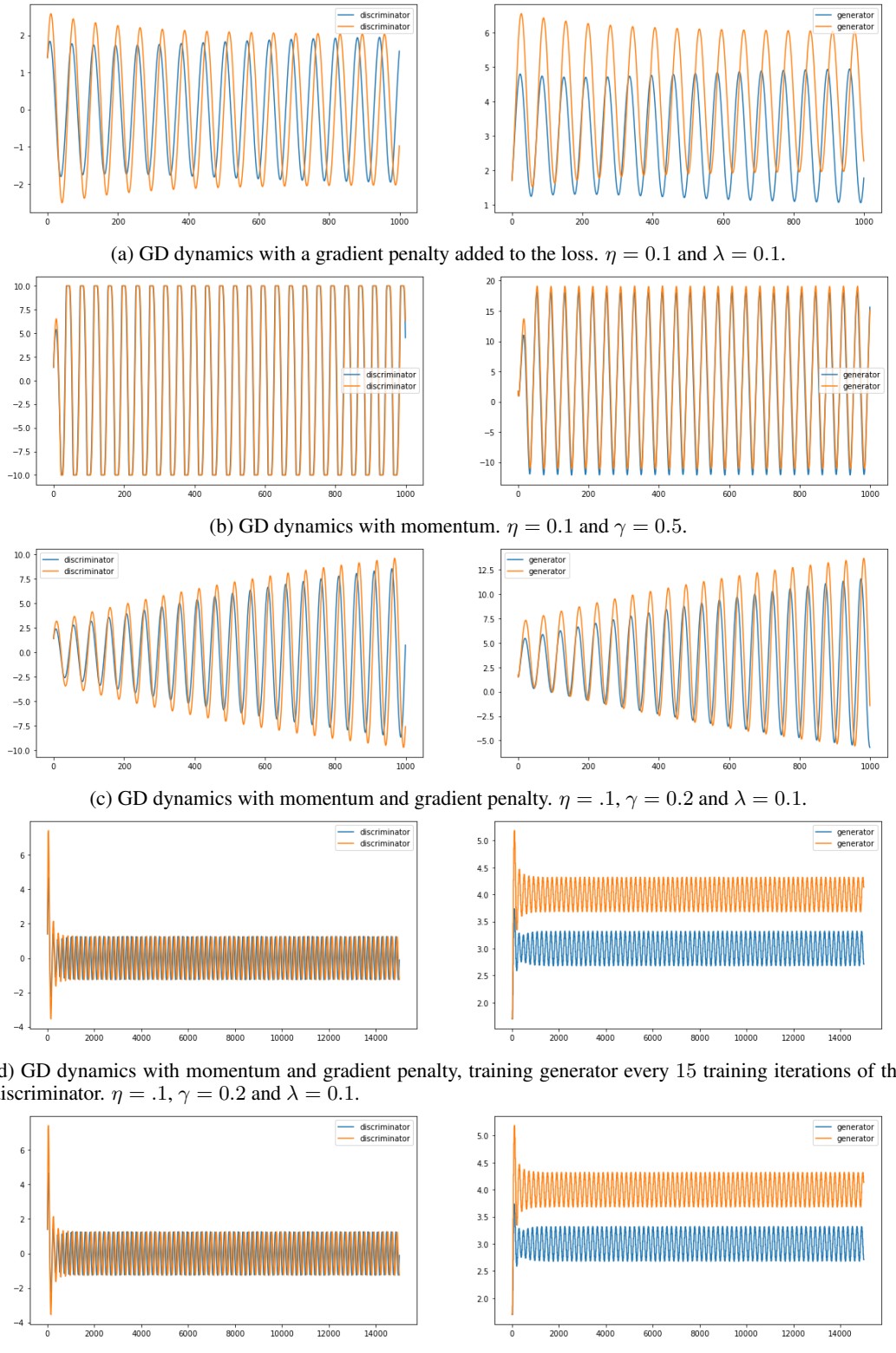

(a) GD dynamics with a gradient penalty added to the loss. $\eta = 0.1$ and $\lambda = 0.1$.

(b) GD dynamics with momentum. $\eta = 0.1$ and $\gamma = 0.5$.

(c) GD dynamics with momentum and gradient penalty. $\eta = .1$, $\gamma = 0.2$ and $\lambda = 0.1$.

(d) GD dynamics with momentum and gradient penalty, training generator every 15 training iterations of the discriminator. $\eta = .1$, $\gamma = 0.2$ and $\lambda = 0.1$.

(e) GD dynamics with Nesterov momentum and gradient penalty, training generator every 15 training iterations of the discriminator. $\eta = .1$, $\gamma = 0.2$ and $\lambda = 0.1$.

Figure 5: Persistence of limit cycles in multiple variants of GD training.

## C    ANOTHER EXAMPLE: LEARNING A CO-VARIANCE MATRIX

We demonstrate the benefits of using OMD over GD in another simple illustrative example. In this case, the example is does not boil down to a bi-linear game and therefore, the simulation results portray that the theoretical results we provided for bi-linear games, carry over qualitatively beyond the linear case.

Consider the case where the data distribution is a mean zero multi-variate normal with an unknown co-variance matrix, i.e., $x \sim N(0, \Sigma)$. We will consider the case where the discriminator is the set of all quadratic functions:

$$D_W(x) = \sum_{ij} W_{ij} x_i x_j = x^T W x \tag{24}$$

The generator is a linear function of the random input noise $z \sim N(0, I)$, of the form:

$$G_V(z) = Vz \tag{25}$$

The parameters $W$ and $V$ are both $d \times d$ matrices. The WGAN game loss associated with these functions is then:

$$L(V, W) = \mathbb{E}_{x \sim N(0,\Sigma)} \left[ x^T W x \right] - \mathbb{E}_{z \sim N(0,I)} \left[ z^T V^T W V z \right] \tag{26}$$

Expanding the latter we get:

$$
\begin{aligned}
L(V, W) =& \mathbb{E}_{x \sim N(0,\Sigma)} \left[ \sum_{ij} W_{ij} x_i x_j \right] - \mathbb{E}_{z \sim N(0,I)} \left[ \sum_{ij} W_{ij} \sum_k V_{ik} z_k \sum_m V_{jm} z_m \right] \\
=& \mathbb{E}_{x \sim N(0,\Sigma)} \left[ \sum_{ij} W_{ij} x_i x_j \right] - \mathbb{E}_{z \sim N(0,I)} \left[ \sum_{ijkm} W_{ij} V_{ik} V_{jm} z_k z_m \right] \\
=& \sum_{ij} W_{ij} \mathbb{E}_{x \sim N(0,\Sigma)} \left[ x_i x_j \right] - \sum_{ijkm} W_{ij} V_{ik} V_{jm} \mathbb{E}_{z \sim N(0,I)} \left[ z_k z_m \right] \\
=& \sum_{ij} W_{ij} \Sigma_{ij} - \sum_{ijkm} W_{ij} V_{ik} V_{jm} 1\{k = m\} \\
=& \sum_{ij} W_{ij} \Sigma_{ij} - \sum_{ijk} W_{ij} V_{ik} V_{jk} \\
=& \sum_{ij} W_{ij} \left( \Sigma_{ij} - \sum_k V_{ik} V_{jk} \right)
\end{aligned}
$$

Given that the covariance matrix is symmetric positive definite, we can write it as $\Sigma = UU^T$. Then the loss simplifies to:

$$L(V, W) = \sum_{ij} W_{ij} \left( \Sigma_{ij} - \sum_k V_{ik} V_{jk} \right) = \sum_{ijk} W_{ij} \left( U_{ik} U_{jk} - V_{ik} V_{jk} \right) \tag{27}$$

The equilibrium of this game is for the generator to choose $V_{ik} = U_{ik}$ for all $i, k$, and for the discriminator to pick $W_{ij} = 0$. For instance, in the case of a single dimension we have $L(V, W) = W \cdot (\sigma^2 - V^2)$, where $\sigma^2$ is the variance of the Gaussian. Hence, the equilibrium is for the generator to pick $V = \sigma$ and the discriminator to pick $W = 0$.

**Dynamics without sampling noise.**    For the mean GD dynamics the update rules are as follows:

$$
\begin{aligned}
W_{ij}^t =& W_{ij}^{t-1} + \eta \left( \Sigma_{ij} - \sum_k V_{ik}^{t-1} V_{jk}^{t-1} \right) \\
V_{ij}^t =& V_{ij}^{t-1} + \eta \sum_k \left( W_{ik}^{t-1} + W_{ki}^{t-1} \right) V_{kj}^{t-1}
\end{aligned}
\tag{28}
$$

We can write the latter updates in a simpler matrix form:

$$W_t = W_{t-1} + \eta \left( \Sigma - V_{t-1}V_{t-1}^T \right)$$
$$V_t = V_{t-1} + \eta(W_{t-1} + W_{t-1}^T)V_{t-1}$$

(GD for Covariance)

Similarly the OMD dynamics are:

$$W_t = W_{t-1} + 2\eta \left( \Sigma - V_{t-1}V_{t-1}^T \right) - \eta \left( \Sigma - V_{t-2}V_{t-2}^T \right)$$
$$V_t = V_{t-1} + 2\eta(W_{t-1} + W_{t-1}^T)V_{t-1} - \eta(W_{t-2} + W_{t-2}^T)V_{t-2}$$

(OMD for Covariance)

Due to the non-convexity of the generators problem and because there might be multiple optimal solutions (e.g. if $\Sigma$ is not strictly positive definite), it is helpful in this setting to also help dynamics by adding $\ell_2$ regularization to the loss of the game. The latter simply adds an extra $2\lambda W_t$ at each gradient term $\nabla_W L(V_t, W_t)$ for the discriminator and a $2\lambda V_t$ at each gradient term $\nabla_V L(V_t, W_t)$ for the generator. In Figures 7 and 6 we give the weights and the implied covariance matrix $\Sigma^G = VV^T$ of the generator's distribution for each of the dynamics for an example setting of the step-size and regularization parameters and for two and three dimensional gaussians respectively. We again see how OMD can stabilize the dynamics to converge pointwise.

**Stochastic dynamics.** In Figure 8 and 9 we also portray the instability of GD and the robustness of the stability of OMD under stochastic dynamics. In the case of stochastic dynamics the gradients are replaced with unbiased estimates or with averages of unbiased estimates over a small minibatch. In the case of a mini-batch of one, the unbiased estimates of the gradients in this setting take the following form:

$$\hat{\nabla}_{W,t} = x_t x_t^T - V_t z_t z_t^T V_t^T$$
$$\hat{\nabla}_{V,t} = -(W_t + W_t^T)V_t z_t z_t^T$$

(Stochastic Gradients)

where $x_t, z_t$ are samples drawn from the true distribution and from the random noise distribution respectively. Hence, the stochastic dynamics simply follow by replacing gradients with unbiased estimates:

$$W_t = W_{t-1} + \eta \hat{\nabla}_{W,t-1}$$
$$V_t = V_{t-1} - \eta \hat{\nabla}_{V,t-1}$$

(SGD for Covariance)

$$W_t = W_{t-1} + 2\eta \hat{\nabla}_{W,t-1} - \eta \hat{\nabla}_{W,t-2}$$
$$V_t = V_{t-1} - 2\eta \hat{\nabla}_{V,t-1} + \eta \hat{\nabla}_{V,t-2}$$

(SOMD for Covariance)

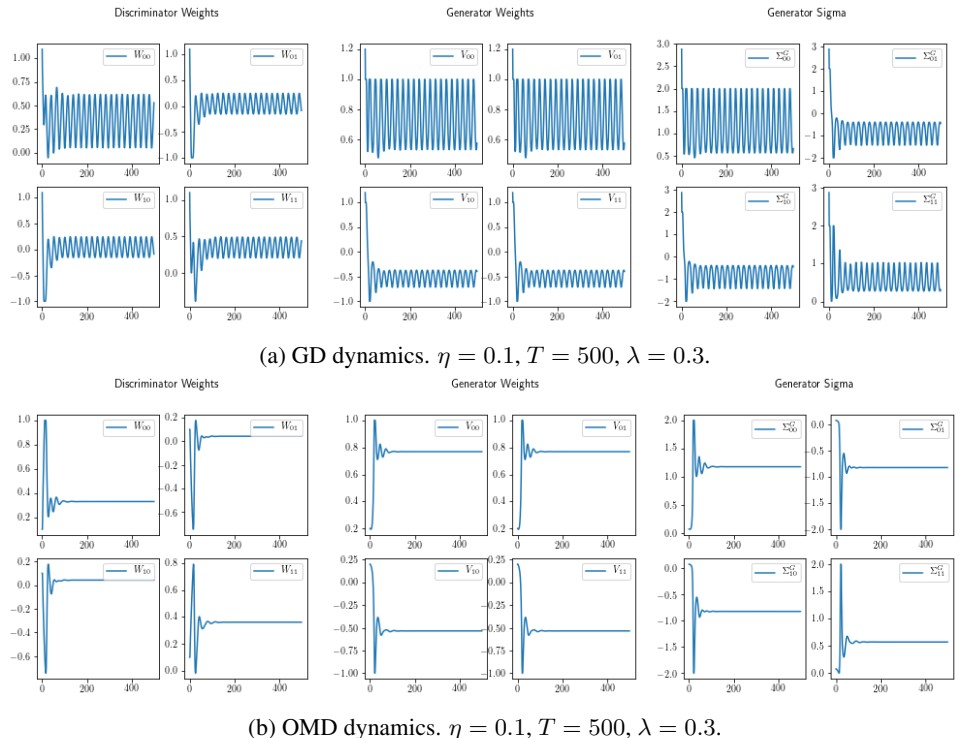

(a) GD dynamics. $\eta = 0.1$, $T = 500$, $\lambda = 0.3$.

(b) OMD dynamics. $\eta = 0.1$, $T = 500$, $\lambda = 0.3$.

Figure 6: Stability of OMD vs GD in the co-variance learning problem for a two-dimensional gaussian ($d = 2$). Weight clipping in $[-1, 1]$ was applied in both dynamics.

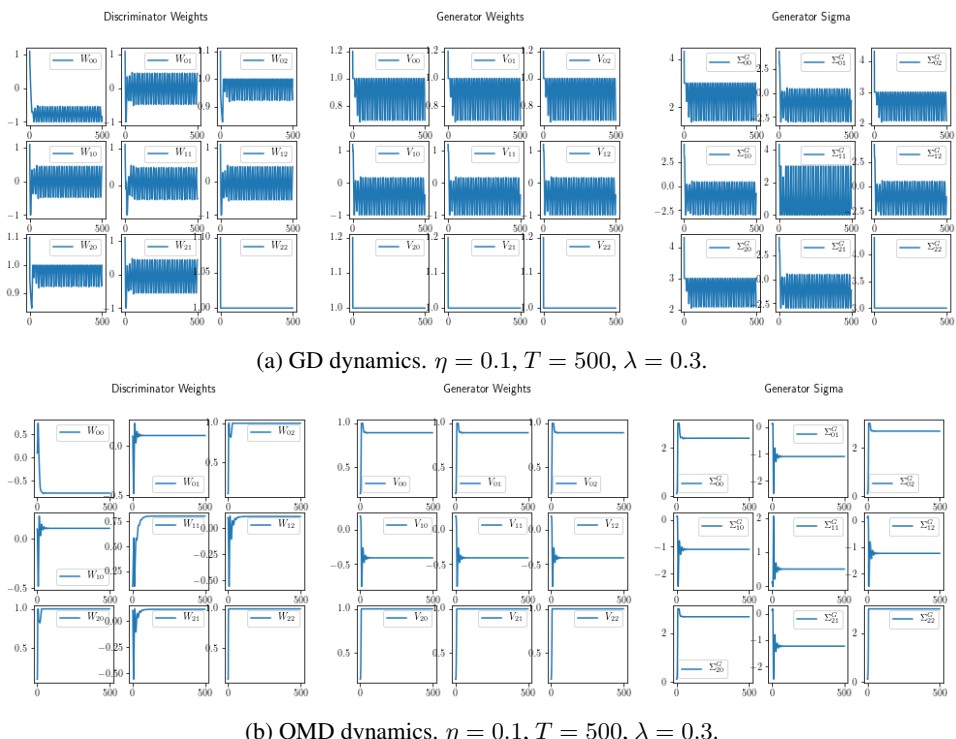

(a) GD dynamics. $\eta = 0.1$, $T = 500$, $\lambda = 0.3$.

(b) OMD dynamics. $\eta = 0.1$, $T = 500$, $\lambda = 0.3$.

Figure 7: Stability of OMD vs GD in the co-variance learning problem for a three-dimensional gaussian ($d = 3$). Weight clipping in $[-1, 1]$ was applied in both dynamics.

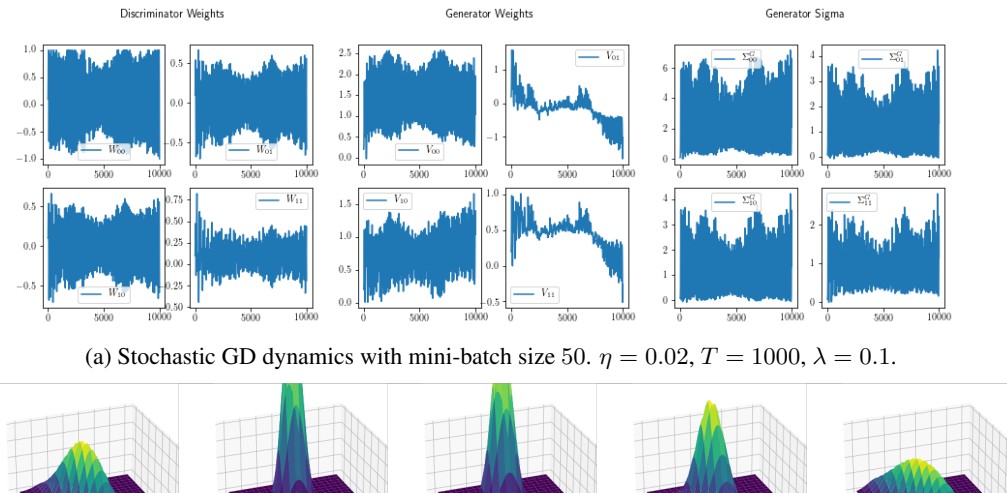

(a) Stochastic GD dynamics with mini-batch size 50. $\eta = 0.02$, $T = 1000$, $\lambda = 0.1$.

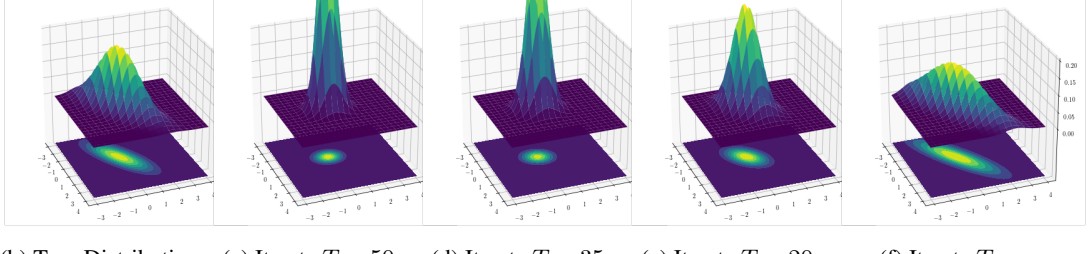

(b) True Distribution    (c) Iterate $T - 50$    (d) Iterate $T - 35$    (e) Iterate $T - 20$    (f) Iterate $T$

(g) Comparison of true distribution and distribution of generator at various points closer to the end of training.

Figure 8: Stochastic GD dynamics for covariance learning of a two-dimensional gaussian ($d = 2$). Weight clipping in $[-1, 1]$ was applied to the discriminator weights.

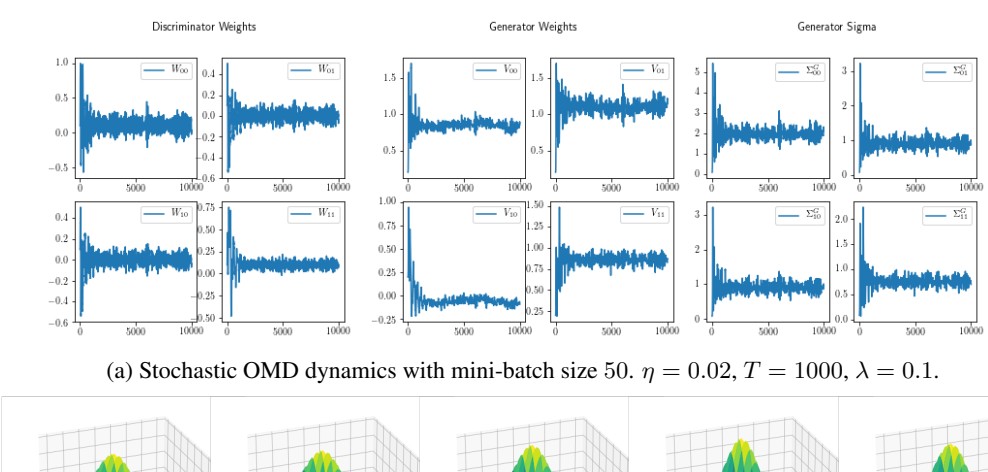

(a) Stochastic OMD dynamics with mini-batch size 50. $\eta = 0.02$, $T = 1000$, $\lambda = 0.1$.

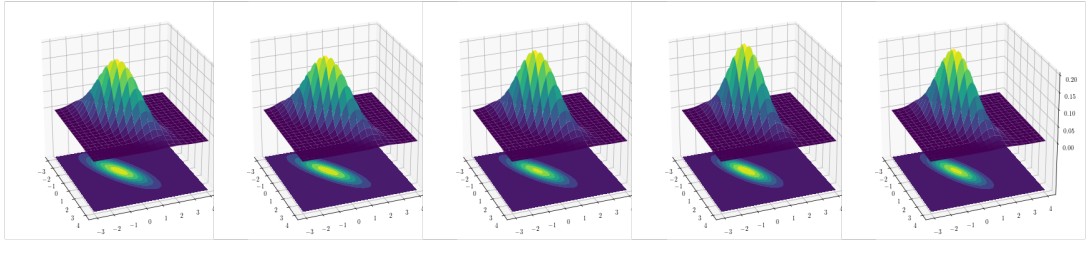

(b) True Distribution    (c) Iterate $T - 50$    (d) Iterate $T - 35$    (e) Iterate $T - 20$    (f) Iterate $T$

(g) Comparison of true distribution and distribution of generator at various points closer to the end of training.

Figure 9: Stability of OMD with stochastic gradients for covariance learning of a two-dimensional gaussian ($d = 2$). Weight clipping in $[-1, 1]$ was applied to the discriminator weights.

# D  LAST ITERATE CONVERGENCE OF OMD IN BILINEAR CASE

The goal of this section is to show that Optimistic Mirror Descent exhibits last iterate convergence to min-max solutions for bilinear functions. In Section D.1, we provide the proof of Theorem 1, that OMD exhibits last iterate convergence to min-max solutions of the following min-max problem

$$\min_x \max_y x^T A y, \tag{29}$$

where $A$ is an abitrary matrix and $x$ and $y$ are unconstrained. In Section D.2, we state the appropriate extension of our theorem to the general case:

$$\inf_x \sup_y \left( x^T A y + b^T x + c^T y + d \right). \tag{30}$$

## D.1  PROOF OF THEOREM 1

As stated in Section 4, for the min-max problem (29) Optimistic Mirror Descent takes the following form, for all $t \geq 1$:

$$x_t = x_{t-1} - 2\eta A y_{t-1} + \eta A y_{t-2} \tag{31}$$

$$y_t = y_{t-1} + 2\eta A^T x_{t-1} - \eta A^T x_{t-2} \tag{32}$$

where for the above iterations to be meaningful we need to specify $x_0, x_{-1}, y_0, y_{-1}$.

As stated in Section 4 we allow any initialization $x_0 \in \mathcal{R}(A)$, and $y_0 \in \mathcal{R}(A^T)$, and set $x_{-1} = 2x_0$ and $y_{-1} = 2y_0$, where $\mathcal{R}(\cdot)$ represents column space. In particular, our initialization means that the first step taken by the dynamics gives $x_1 = x_0$ and $y_1 = y_0$.

Before giving our proof of Theorem 1, we need some further notation. For all $i \in \mathbb{N}$, we set:

$$M_i = A^j (A^T A)^k, N_i = \left( A^T \right)^j \left( A A^T \right)^k$$

$$\Delta_t^i = ||N_i A y_t||_2^2 + ||M_i A^T x_t||_2^2$$

where $k \in \mathbb{Z}$ and $j \in \{0, 1\}$ are such that: $i = 2k + j$.

With this notation, $\Delta_t^0 = ||A^T x_t||_2^2 + ||A y_t||_2^2$, $\Delta_t^1 = ||A A^T x_t||_2^2 + ||A^T A y_t||_2^2$, $\Delta_t^2 = ||A^T A A^T x_t||_2^2 + ||A A^T A y_t||_2^2$, etc.

We also use the notation $\langle u, v \rangle_X = u^T X X^T v$, for vectors $u, v \in \mathbb{R}^d$ and square $d \times d$ matrices $X$. We similarly define the norm notation $||u||_X = \sqrt{\langle u, u \rangle_X}$. Given our notation, we have the following claim, shown in Appendix D.3.

**Claim 1.** *For all matrices $A$ and vectors $u, v$ of the appropriate dimensions:*

$$\langle Au, Av \rangle_{AM_i^T} = \langle u, v \rangle_{A^T N_{i+1}^T}; \quad \langle A^T u, A^T v \rangle_{A^T N_i^T} = \langle u, v \rangle_{AM_{i+1}^T}; \quad \langle u, Av \rangle_{AM_i^T} = \langle v, A^T u \rangle_{A^T N_i^T}.$$

With our notation in place, we show (through iterated expansion of the update rule), the following lemma, proved in Appendix D.3:

**Lemma 2.** *For the dynamics of Eq. (31) and (32) and any initialization $\frac{1}{2}x_{-1} = x_0 \in \mathcal{R}(A)$, and $\frac{1}{2}y_{-1} = y_0 \in \mathcal{R}(A^T)$ we have the following for all $i, t \in \mathbb{N}$ such that $i \geq 0$ and $t \geq 2$:*

$$\Delta_t^i - \Delta_{t-1}^i = 4\eta^2 \Delta_{t-1}^{i+1} - 5\eta^2 \Delta_{t-2}^{i+1} - 2\eta^3 \left( \langle x_{t-2}, A y_{t-4} \rangle_{AM_{i+1}^T} - \langle y_{t-2}, A^T x_{t-4} \rangle_{A^T N_{i+1}^T} \right).$$

We are ready to prove Theorem 1. Its proof is implied by the following stronger theorem, and Corollary 7.

**Theorem 3.** *Consider the dynamics of Eq. (31) and (32) and any initialization $\frac{1}{2}x_{-1} = x_0 \in \mathcal{R}(A)$, and $\frac{1}{2}y_{-1} = y_0 \in \mathcal{R}(A^T)$. Let*

$$\gamma = \max \left( \left|\left| \left( A A^T \right)^+ \right|\right|, \left|\left| \left( A^T A \right)^+ \right|\right| \right),$$

*where for a matrix $X$ we denote by $X^+$ its generalized inverse and by $||X||$ its spectral norm. Suppose that $\max\{||A||, ||A^T||\} \equiv \lambda_\infty \leq 1$ and $\eta$ is a small enough constant satisfying $\eta < 1/(3\gamma^2)$. Then, for all $i \in \mathbb{N}$:*

$$\Delta_1^i = \Delta_0^i, \tag{33}$$

$$\Delta_2^i \leq (1+\eta)^2 \Delta_0^i, \tag{34}$$

*and, for all $i, t \in \mathbb{N}$ such that $t \geq 3$, the following condition holds:*

$$H(i,t): \quad \Delta_t^i \leq \left(1 - \frac{\eta^2}{\gamma^2}\right) \Delta_{t-1}^i + 16\eta^3 \Delta_0^0. \tag{35}$$

*Proof.* Eq. (33) holds trivially as under our initialization $x_1 = x_0$ and $y_1 = y_0$. Eq. (34) is also easy to show by noticing the following. Given our initialization:

$$x_2 = x_0 - \eta A y_0$$
$$y_2 = y_0 + \eta A x_0$$

Hence (using $j = i \mod 2$):

$$M_i A^T x_2 = M_i A^T x_0 - \eta M_i A^T A y_0 \tag{36}$$

$$\Rightarrow \quad ||M_i A^T x_2||_2 \leq ||M_i A^T x_0||_2 + \eta ||M_i A^T A y_0||_2 \tag{37}$$

$$= ||M_i A^T x_0||_2 + \eta ||A^j (A^T)^{1-j} N_i A y_0||_2 \tag{38}$$

$$\leq ||M_i A^T x_0||_2 + \eta \lambda_\infty ||N_i A y_0||_2 \tag{39}$$

$$\leq ||M_i A^T x_0||_2 + \eta ||N_i A y_0||_2 \tag{40}$$

Similarly:

$$||N_i A y_2||_2 \leq ||N_i A y_0||_2 + \eta ||M_i A^T x_0||_2 \tag{41}$$

It follows from (40) and (41) that

$$\Delta_2^i \leq (1+\eta)^2 \Delta_0^i.$$

We use induction on $t$ to prove (35). We start our proof by showing the inductive step, and postpone establishing the basis of our induction to the end of this proof. For the inductive step, we assume that $H(i, \tau)$ holds for all $i \geq 0$ and $1 \leq \tau < t$, for some $t > 3$. Assuming this, we show next that $H(i, t)$ holds for all $i$. To do this, we make use of a few lemmas, whose proofs are given in Appendix D.3.

**Lemma 4.** *Under the conditions of the theorem, for all $i \geq 0, t \geq 2$:*

$$\Delta_t^i - \Delta_{t-1}^i \leq 4\eta^2 \Delta_{t-1}^{i+1} - 5\eta^2 \Delta_{t-2}^{i+1} + \eta^3 (\Delta_{t-2}^{i+1} + \Delta_{t-4}^{i+1}).$$

**Lemma 5.** *Under the conditions of the theorem, for all $i, t \geq 0$: $\Delta_t^{i+1} \leq \Delta_t^i$.*

**Lemma 6.** *Under the conditions of the theorem, for all $i \geq 0, t \geq 0$:*

$$\Delta_t^{i+2} \geq \frac{1}{\gamma^2} \Delta_t^i.$$

Given these lemmas, we show our inductive step. So for $t \geq 4$:

$$\Delta_t^i - \Delta_{t-1}^i \leq 4\eta^2 \Delta_{t-1}^{i+1} - 5\eta^2 \Delta_{t-2}^{i+1} + \eta^3(\Delta_{t-2}^{i+1} + \Delta_{t-4}^{i+1}) \tag{42}$$

$$\leq -\eta^2 \Delta_{t-1}^{i+1} + \eta^3(\Delta_{t-2}^{i+1} + \Delta_{t-4}^{i+1}) + 80\eta^5 \Delta_0^0 \tag{43}$$

$$\leq -\frac{1}{\gamma^2}\eta^2 \Delta_{t-1}^{i-1} + \eta^3(\Delta_{t-2}^{i+1} + \Delta_{t-4}^{i+1}) + 80\eta^5 \Delta_0^0 \tag{44}$$

$$\leq -\frac{1}{\gamma^2}\eta^2 \Delta_{t-1}^{i-1} + \eta^3(\Delta_{t-2}^0 + \Delta_{t-4}^0) + 80\eta^5 \Delta_0^0 \tag{45}$$

$$\leq -\frac{1}{\gamma^2}\eta^2 \Delta_{t-1}^{i-1} + \left(2\eta^3 \Delta_2^0 + 2\eta^3 \frac{\gamma^2}{\eta^2} 16\eta^3 \Delta_0^0\right) + 80\eta^5 \Delta_0^0 \tag{46}$$

$$\leq -\frac{1}{\gamma^2}\eta^2 \Delta_{t-1}^{i-1} + \left(2\eta^3(1+\eta)^2 + 32\eta^3 \frac{\gamma^2}{\eta^2}\eta^3 + 80\eta^5\right)\Delta_0^0 \tag{47}$$

$$\leq -\frac{1}{\gamma^2}\eta^2 \Delta_{t-1}^{i-1} + \left(2\eta^3(1+\eta)^2 + 11\eta^3 + 80\eta^5\right)\Delta_0^0 \tag{48}$$

$$\leq -\frac{1}{\gamma^2}\eta^2 \Delta_{t-1}^{i-1} + 16\eta^3 \Delta_0^0 \tag{49}$$

$$\leq -\frac{1}{\gamma^2}\eta^2 \Delta_{t-1}^i + 16\eta^3 \Delta_0^0 \tag{50}$$

where for the first inequality we used Lemma 4, for the second inequality we used that $\Delta_{t-1}^{i+1} \leq \Delta_{t-2}^{i+1} + 16\eta^3 \Delta_0^0$ (which is implied by the induction hypothesis), for the third inequality we used Lemma 6, for the fourth inequality we used Lemma 5, for the fifth inequality we applied the induction hypothesis iteratively, for the sixth inequality we used Eq. (34), for the seventh and eighth inequality we used that $\eta$ is small enough, and for the last inequality we used Lemma 5. Hence:

$$\Delta_t^i \leq \left(1 - \frac{\eta^2}{\gamma^2}\right)\Delta_{t-1}^i + 16\eta^3 \Delta_0^0.$$

This completes the proof of our inductive step.

It remains to show the basis of the induction, namely that $H(i,3)$ holds for all $i \in \mathbb{N}$. From Lemma 4 we have:

$$\Delta_3^i - \Delta_2^i \leq 4\eta^2 \Delta_2^{i+1} - 5\eta^2 \Delta_1^{i+1} + \eta^3(\Delta_1^{i+1} + \Delta_{-1}^{i+1}) \tag{51}$$

$$\leq 4\eta^2 \Delta_2^{i+1} - 5\eta^2 \Delta_0^{i+1} + 5\eta^3 \Delta_0^{i+1} \tag{52}$$

$$= -\eta^2 \Delta_2^{i+1} + 5\eta^2(\Delta_2^{i+1} - \Delta_0^{i+1}) + 5\eta^3 \Delta_0^{i+1} \tag{53}$$

$$\leq -\eta^2 \Delta_2^{i+1} + 5\eta^3(2+\eta)\Delta_0^{i+1} + 5\eta^3 \Delta_0^{i+1} \tag{54}$$

$$= -\eta^2 \Delta_2^{i+1} + 5\eta^3(3+\eta)\Delta_0^{i+1} \tag{55}$$

$$\leq -\eta^2 \Delta_2^{i+1} + 15\eta^3(1+\eta/3)\Delta_0^0 \tag{56}$$

$$\leq -\frac{\eta^2}{\gamma^2}\Delta_2^{i-1} + 15\eta^3(1+\eta/3)\Delta_0^0 \tag{57}$$

$$\leq -\frac{\eta^2}{\gamma^2}\Delta_2^i + 15\eta^3(1+\eta/3)\Delta_0^0, \tag{58}$$

where for the second equality we used that $0.5x_{-1} = x_0 = x_1$ and $0.5y_{-1} = y_0 = y_1$ (which follow from our initialization), for the third inequality we used that (34), for the fourth inequality we used Lemma 5, for the fifth inequality we used Lemma 6, and for the last inequality we used Lemma 5. Hence, for small enough $\eta$, we have:

$$\Delta_3^i \leq \left(1 - \frac{\eta^2}{\gamma^2}\right)\Delta_2^i + 16\eta^3 \Delta_0^0.$$

$\square$

**Corollary 7.** *Under the conditions of Theorem 3, $\Delta_t^0 \equiv \left\|A^T x_t\right\|_2^2 + \left\|Ay_t\right\|_2^2 \to O(\eta\gamma^2\Delta_0^0)$ as $t \to +\infty$. In particular, for large enough $t$, the last iterate of OMD is within $O\left(\sqrt{\eta}\cdot\gamma\sqrt{\Delta_0^0}\right)$ distance from the space of equilibrium points of the game, where $\sqrt{\Delta_0^0}$ is the distance of the initial point $(x_0, y_0)$ from the equilibrium space, and where both distances are taken with respect to the norm $\sqrt{x^T AA^T x + y^T A^T Ay}$.*

*Proof of Corollary 7:* It follows from (33), (34) and (35) that:

$$\Delta_t^0 \leq \left(1 - \frac{\eta^2}{\gamma^2}\right)^{t-2}(1+\eta)^2\Delta_0^0 + 16\sum_{t=0}^{\infty}\left(1 - \frac{\eta^2}{\gamma^2}\right)^t \eta^3\Delta_0^0$$

$$= \left(1 - \frac{\eta^2}{\gamma^2}\right)^{t-2}(1+\eta)^2\Delta_0^0 + O\left(\eta\gamma^2\Delta_0^0\right),$$

which shows the first part of our claim. For the second part of our claim recall that the solutions to (29) are all pairs $(x, y)$ such that $x$ is in the null space of $A^T$ and $y$ is in the null space of $A$. □

## D.2 GENERAL BILINEAR CASE

**Theorem 8.** *Consider OMD for the min-max problem (30):*

$$\inf_x \sup_y \left(x^T Ay + b^T x + c^T y + d\right).$$

*Under the same conditions as Corollary 7 and whenever (30) is finite, OMD exhibits last iterate convergence in the same sense as in Corollary 7. In particular, for large enough $t$, the last iterate of OMD is within $O\left(\sqrt{\eta}\cdot\gamma\sqrt{\Delta_0^0}\right)$ distance from the space of equilibrium points of the game, where $\sqrt{\Delta_0}$ is the distance of the point $(x_0 + (A^T)^+c, y_0 + A^+b)$ from the equilibrium space, and where both distances are taken with respect to the norm $\sqrt{x^T AA^T x + y^T A^T Ay}$. Whenever (30) is infinite or undefined, the OMD dynamics travels to infinity and we characterize its motion.*

*Proof of Theorem 8:* Trivially, we need only consider functions of the form $x^T Ay + b^T x + c^T y$. We consider the following decompositions of $b$ and $c$:

$$b = b_1 + b_2 \qquad\qquad \text{where } b_1 \in \mathcal{R}(A), b_2 \in \mathcal{N}(A^T)$$

$$c = c_1 + c_2 \qquad\qquad \text{where } c_1 \in \mathcal{R}(A^T), c_2 \in \mathcal{N}(A)$$

Given the above we can also define $b_3$ and $c_3$ as follows:

$$Ac_3 = b_1 \qquad\qquad \text{feasible since } b_1 \in \mathcal{R}(A)$$

$$A^T b_3 = c_1 \qquad\qquad \text{feasible since } c_1 \in \mathcal{R}(A^T)$$

Then, we can make the following variable substition:

$$\alpha_t = x_t + \eta t b_2 + b_3$$

$$\beta_t = y_t - \eta t c_2 + c_3$$

so that:

$$A^T \alpha_t = A^T x_t + \eta t A^T b_2 + A^T b_3$$

$$= A^T x_t + c_1 \quad \text{since } b_2 \in \mathcal{N}(A^T)$$

$$A\beta_t = Ay_t - \eta t Ac_2 + Ac_3$$

$$= Ay_t + b_1 \quad \text{since } c_2 \in \mathcal{N}(A)$$

We also state the OMD dynamics for $x_t$ and $y_t$ for problem (30):

$$x_t = x_{t-1} - 2\eta(Ay_{t-1} + b) + \eta(Ay_{t-2} + b)$$

$$= x_{t-1} - 2\eta Ay_{t-1} + \eta Ay_{t-2} - \eta b$$

$$y_t = y_{t-1} + 2\eta(A^T x_{t-1} + c) - \eta(A^T x_{t-2} + c)$$

$$= y_{t-1} + 2\eta A^T x_{t-1} - \eta A^T x_{t-2} + \eta c$$

Note that given this update step:

$$x_{t+1} = x_t - 2\eta A y_t + \eta A y_{t-1} - \eta b$$
$$x_{t+1} = x_t - \eta b_2 - 2\eta A y_t + \eta A y_{t-1} - \eta A c_3$$
$$x_{t+1} = x_t - \eta b_2 - 2\eta A(y_t + c_3) + \eta A(y_{t-1} + c_3)$$
$$x_{t+1} = x_t - \eta b_2 - 2\eta A(y_t - \eta c_2 t + c_3) + \eta A(y_{t-1} - \eta c_2(t-1) + c_3)$$
$$x_{t+1} + \eta b_2(t+1) = x_t + \eta b_2 t - 2\eta A(y_t - \eta c_2 t + c_3) + \eta A(y_{t-1} - \eta c_2(t-1) + c_3)$$
$$x_{t+1} + \eta b_2(t+1) + b_3 = x_t + \eta b_2 t + b_3 - 2\eta A(y_t - \eta c_2 t + c_3) + \eta A(y_{t-1} - \eta c_2(t-1) + c_3)$$
$$\alpha_{t+1} = \alpha_t - 2\eta A \beta_t + \eta A \beta_{t-1}$$

Analogously:

$$\beta_{t+1} = \beta_t + 2\eta A^T \alpha_t - \eta A^T \alpha_{t-1}$$

Note that these are precisely the dynamics for which we proved convergence in Theorem 1. Thus, by invoking Theorem 3 and Corollary 7 on the sequence $(\alpha_t, \beta_t)$ and then substituting back $(x_t, y_t)$, we have that for all large enough $t$:

$$x_t = -\eta b_2 t - b_3 + \epsilon_x(t)$$
$$y_t = \eta c_2 t - c_3 + \epsilon_y(t)$$

$$\text{such that } ||A^T \epsilon_x(t)||_2, ||A \epsilon_y(t)||_2 \in O\left(\sqrt{\eta} \cdot \gamma \sqrt{\Delta_0^0}\right),$$

where $\Delta_0^0 = ||A^T(x_0 + b_3)||_2^2 + ||A(y_0 + c3)||_2^2$.

In particular, this shows that, whenever (30) is finite (i.e. $b_2 = c_2 = 0$), OMD exhibits last iterate convergence. For large enough $t$, the last iterate of OMD is within $O\left(\sqrt{\eta} \cdot \gamma \sqrt{\Delta_0^0}\right)$ distance from the space of equilibrium points of the game, where $\sqrt{\Delta_0^0}$ is the distance of $(x_0 + b_3, y_0 + c_3)$ from the equilibrium space in the norm $\sqrt{x^T A A^T x + y^T A^T A y}$. Whenever (30) is infinite or undefined, the OMD dynamics travels to infinity linearly, with fluctuations around the divergence specified as above. □

## D.3 OMITTED PROOFS

*Proof of Proposition 1:* To show this, we consider the $\ell_2$ distance of the solution at time $t$. First, recall the GD update step in the special case of $f(x, y) = x^T y$:

$$x_t = x_{t-1} - \eta y_{t-1}$$
$$y_t = y_{t-1} + \eta x_{t-1}$$

Then, note that the squared $\ell_2$ distance of the running iterate $(x_t, y_t)$ to the unique equilibrium solution $(0, 0)$ is given by $d(t) := ||x_t||_2^2 + ||y_t||_2^2$, which we can calculate:

$$||x_t||_2^2 = ||x_{t-1}||_2^2 - 2\eta x_{t-1}^T y_{t-1} + \eta^2 ||y_{t-1}||_2^2$$
$$||y_{t-1}||_2^2 = ||y_{t-1}||_2^2 + 2\eta y_{t-1}^T x_{t-1} + \eta^2 ||x_{t-1}||_2^2$$
$$\therefore d(t) = d(t-1) + \eta^2 d(t-1)$$
$$= (1 + \eta^2)d(t-1)$$

This indicates that for any value of $\eta > 0$, the running iterate of GD *diverges* from the equilibrium. □

*Proof of Claim 1:* For our first claim, observe that:

$$\langle Au, Av \rangle_{AM_i^T} = u^T A^T A M_i^T M_i A^T A v$$
$$= u^T A^T A(A^T A)^k (A^j)^T A^j (A^T A)^k A^T A v$$
$$= u^T A^T (A A^T)^k A(A^j)^T A^j A^T (A A^T)^k A v$$
$$= u^T A^T [(A A^T)^k A(A^j)^T][A^j A^T (A A^T)^k] A v$$
$$= u^T A^T N_{i+1}^T N_{i+1} A v$$
$$= \langle u, v \rangle_{A^T N_{i+1}^T}$$

Our second claim, $\langle A^T u, A^T v \rangle_{A^T N_i^T} = \langle u, v \rangle_{AM_{i+1}^T}$, is proven analogously.

For our third claim:

$$
\begin{aligned}
\langle u, Av \rangle_{AM_i^T} &= u^T A M_i^T M_i A^T A v \\
&= u^T A (A^T A)^k (A^T A)^j (A^T A)^k A^T A v \\
\text{if } j = 0: \\
&= u^T A (A^T A)^k (A^T A)^k A^T A v \\
&= u^T A [A^T (AA^T)^k][(AA^T)^k A]v \\
&= u^T A [A^T N_i^T][N_i A]v \\
&= \langle v, A^T u \rangle_{A^T N_i^T} \\
\text{otherwise:} \\
&= u^T A (A^T A)^k A^T A (A^T A)^k A^T A v \\
&= u^T A [(A^T A)^k A^T A][(A^T A)^k A^T A]v \\
&= u^T A [A^T (AA^T)^k A][A^T (AA^T)^k A]v \\
&= u^T A [A^T N_i^T][N_i A]v \\
&= \langle v, A^T u \rangle_{A^T N_i^T}
\end{aligned}
$$

$\square$

*Proof of Lemma 2:* First, we note the following scaled update rule:

$$
M_i A^T x_t = M_i \left( A^T x_{t-1} - 2\eta A^T A y_{t-1} + \eta A^T A y_{t-2} \right)
$$
$$
N_i A y_t = N_i \left( A y_{t-1} + 2\eta AA^T x_{t-1} - \eta AA^T x_{t-2} \right)
$$

Then, taking the norm of both sides, and using the statements of Claim 1:

$$
\begin{aligned}
||x_t||^2_{AM_i^T} = ||x_{t-1}||^2_{AM_i^T} &+ 4\eta^2 ||y_{t-1}||^2_{A^T N_{i+1}^T} + \eta^2 ||y_{t-2}||^2_{A^T N_{i+1}^T} - 4\eta \langle x_{t-1}, A y_{t-1} \rangle_{AM_i^T} \\
&+ 2\eta \langle x_{t-1}, A y_{t-2} \rangle_{AM_i^T} - 4\eta^2 \langle y_{t-1}, y_{t-2} \rangle_{A^T N_{i+1}^T}
\end{aligned}
$$

$$
\begin{aligned}
||y_t||^2_{A^T N_i^T} = ||y_{t-1}||^2_{A^T N_i^T} &+ 4\eta^2 ||x_{t-1}||^2_{AM_{i+1}^T} + \eta^2 ||x_{t-2}||^2_{AM_{i+1}^T} + 4\eta \langle y_{t-1}, A^T x_{t-1} \rangle_{A^T N_i^T} \\
&+ 2\eta \langle y_{t-1}, A^T x_{t-2} \rangle_{A^T N_i^T} - 4\eta^2 \langle x_{t-1}, x_{t-2} \rangle_{AM_{i+1}^T}
\end{aligned}
$$

$$
\begin{aligned}
\therefore \Delta_t^i &= ||x_t||^2_{AM_i^T} + ||y_t||^2_{A^T N_i^T} \\
&= \Delta_{t-1}^i + 4\eta^2 \Delta_{t-1}^{i+1} + \eta^2 \Delta_{t-2}^{i+1} + 2\eta(\langle x_{t-1}, A y_{t-2} \rangle_{AM_i^T} - \langle y_{t-1}, A x_{t-2} \rangle_{A^T N_i^T}) \\
&\quad - 4\eta^2(\langle x_{t-1}, x_{t-2} \rangle_{AM_{i+1}^T} + \langle y_{t-1}, y_{t-2} \rangle_{A^T N_{i+1}^T})
\end{aligned}
$$

Expanding the first pair of inner products above and using Claim 1 again:

$$
\begin{aligned}
\langle x_{t-1}, A y_{t-2} \rangle_{AM_i^T} - \langle y_{t-1}, A^T x_{t-2} \rangle_{A^T N_i^T} &= \langle x_{t-2} - 2\eta A y_{t-2} + \eta A y_{t-3}, A y_{t-2} \rangle_{AM_i^T} \\
&\quad - \langle y_{t-2} + 2\eta A^T x_{t-2} - \eta A^T x_{t-3}, A^T x_{t-2} \rangle_{A^T N_i^T} \\
&= -2\eta(||y_{t-2}||^2_{A^T N_{i+1}^T} + ||x_{t-2}||^2_{AM_{i+1}^T}) + \eta(\langle x_{t-2}, x_{t-3} \rangle_{AM_{i+1}^T} + \langle y_{t-2}, y_{t-3} \rangle_{A^T N_{i+1}^T})
\end{aligned}
$$

Then, multiplying by $2\eta$ and substituting into the previous derivation yields:

$$
\begin{aligned}
\Delta_t^i - \Delta_{t-1}^i &= 4\eta^2 \Delta_{t-1}^{i+1} - 3\eta^2 \Delta_{t-2}^{i+1} + 2\eta^2(\langle x_{t-2}, x_{t-3} \rangle_{AM_{i+1}^T} + \langle y_{t-2}, y_{t-3} \rangle_{A^T N_{i+1}^T}) \\
&\quad - 4\eta^2(\langle x_{t-1}, x_{t-2} \rangle_{AM_{i+1}^T} + \langle y_{t-1}, y_{t-2} \rangle_{A^T N_{i+1}^T})
\end{aligned}
$$

Now, consider the following inner product:

$$\langle x_{t-2}, x_{t-1} \rangle_{AM_{i+1}^T} + \langle y_{t-2}, y_{t-1} \rangle_{A^T N_{i+1}^T} = \langle x_{t-2}, x_{t-2} - 2\eta A y_{t-2} + \eta A y_{t-3} \rangle_{AM_{i+1}^T}$$

$$+ \langle y_{t-2}, y_{t-2} + 2\eta A^T x_{t-2} - \eta A^T x_{t-3} \rangle_{A^T N_{i+1}^T}$$

$$= \Delta_{t-2}^{i+1} + \eta \left( \langle x_{t-2}, A y_{t-3} \rangle_{AM_{i+1}^T} - \langle y_{t-2}, A^T x_{t-3} \rangle_{A^T N_{i+1}^T} \right)$$

Once again, we multiply by $-4\eta^2$ and substitute:

$$\Delta_t^i - \Delta_{t-1}^i = 4\eta^2 \Delta_{t-1}^{i+1} - 7\eta^2 \Delta_{t-2}^{i+1} + 2\eta^2 (\langle x_{t-2}, x_{t-3} \rangle_{AM_{i+1}^T} + \langle y_{t-2}, y_{t-3} \rangle_{A^T N_{i+1}^T})$$

$$+ 4\eta^3 (\langle y_{t-2}, A^T x_{t-3} \rangle_{A^T N_{i+1}^T} - \langle x_{t-2}, A y_{t-3} \rangle_{AM_{i+1}^T})$$

Now, we use the update step for time $t - 2$. For all $t \geq 1$, this is well-defined, since $x_{-1}$ and $y_{-1}$ are defined. To ensure that this step is sound for $t = 0$ requires we define the following, where $X^+$ denotes the generalized inverse:

$$x_{-2} = 4x_0 + \frac{1}{\eta}(A^T)^+ y_0$$

$$y_{-2} = 4y_0 - \frac{1}{\eta}A^+ x_0$$

We define these such that: $A^T x_{-2} = 4A^T x_0 + \frac{y_0}{\eta}$ and $A y_{-2} = 4A y_0 - \frac{x_0}{\eta}$ (since $x_0 \in R(A)$ and $y_0 \in R(A^T)$), and thus the following equalities hold:

$$x_0 = x_{-1} - 2\eta A y_{-1} + \eta A y_{-2}$$

$$y_0 = y_{-1} + 2\eta A^T x_{-1} - \eta A^T x_{-2}$$

This allows us to use the following expansion freely for all $t \geq 2$:

$$x_{t-2} = x_{t-3} - 2\eta A y_{t-3} + \eta A y_{t-4} \qquad \implies x_{t-3} - 2\eta A y_{t-3} = x_{t-2} - \eta A y_{t-4}$$

$$y_{t-2} = y_{t-3} + 2\eta A^T x_{t-3} - \eta A^T x_{t-4} \qquad \implies y_{t-3} + 2\eta A^T x_{t-3} = y_{y-2} + \eta A^T x_{t-4}$$

We can gather the inner product terms and use this update rule to get our final desired result:

$$\Delta_t^i - \Delta_{t-1}^i = 4\eta^2 \Delta_{t-1}^{i+1} - 7\eta^2 \Delta_{t-2}^{i+1} + 2\eta^2 (\langle x_{t-2}, x_{t-3} - 2\eta A y_{t-3} \rangle_{AM_{i+1}^T} + \langle y_{t-2}, y_{t-3} + 2\eta A^T x_{t-3} \rangle_{A^T N_{i+1}^T})$$

$$= 4\eta^2 \Delta_{t-1}^{i+1} - 5\eta^2 \Delta_{t-2}^{i+1} - 2\eta^3 (\langle x_{t-2}, A y_{t-4} \rangle_{AM_{i+1}^T} - \langle y_{t-2}, A^T x_{t-4} \rangle_{A^T N_{i+1}^T})$$

$$\square$$

*Proof of Lemma 4:* To prove this, first consider the following trivial inequality:

$$\left|\left| y_{t-2} - A^T x_{t-4} \right|\right|_{A^T N_{i+1}^T}^2 + \left|\left| x_{t-2} + A y_{t-4} \right|\right|_{AM_{i+1}^T}^2$$

$$= ||y_{t-2}||_{A^T N_{i+1}^T}^2 - 2\langle y_{t-2}, A^T x_{t-4} \rangle_{A^T N_{i+1}^T} + \left|\left| A^T x_{t-4} \right|\right|_{A^T N_{i+1}^T}^2$$

$$+ ||x_{t-2}||_{AM_{i+1}^T}^2 + 2\langle x_{t-2}, A y_{t-4} \rangle_{AM_{i+1}^T} + ||A y_{t-4}||_{AM_{i+1}^T}^2$$

$$\geq 0$$

Rearranging:

$$2\langle y_{t-2}, A^T x_{t-4} \rangle_{A^T N_{i+1}^T} - 2\langle x_{t-2}, A y_{t-4} \rangle_{AM_{i+1}^T} \leq \Delta_{t-2}^{i+1} + \left( \left|\left| A^T x_{t-4} \right|\right|_{A^T N_{i+1}^T}^2 + ||A y_{t-4}||_{AM_{i+1}^T}^2 \right)$$

$$\leq \Delta_{t-2}^{i+1} + \lambda_\infty^2 \left( ||x_{t-4}||_{AM_{i+1}^T}^2 + ||y_{t-4}||_{A^T N_{i+1}^T}^2 \right)$$

$$\leq \Delta_{t-2}^{i+1} + (\lambda_\infty^2 \Delta_{t-4}^{i+1})$$

$$\leq \Delta_{t-2}^{i+1} + \Delta_{t-4}^{i+1}$$

Now, we can apply this bound to the result of Lemma 2:

$$\Delta_t^i - \Delta_{t-1}^i = 4\eta^2 \Delta_{t-1}^{i+1} - 5\eta^2 \Delta_{t-2}^{i+1} - 2\eta^3 (\langle x_{t-2}, Ay_{t-4} \rangle_{AM_{i+1}^T} - \langle y_{t-2}, A^T x_{t-4} \rangle_{A^T N_{i+1}^T})$$
$$\leq 4\eta^2 \Delta_{t-1}^{i+1} - 5\eta^2 \Delta_{t-2}^{i+1} + \eta^3 (\Delta_{t-2}^{i+1} + \Delta_{t-4}^{i+1})$$

Which is what we sought out to prove. $\qquad\square$

*Proof of Lemma 5:* Suppose $j = i \mod 2$ and $k = (i - j)/2$. Notice the following identities:

$$M_i = A^j (A^T A)^k, N_i = (A^T)^j (AA^T)^k$$
$$M_{i+1} = (A^T)^j A (A^T A)^k, N_{i+1} = A^j A^T (AA^T)^k$$

Now:

$$\Delta_t^{i+1} = ||N_{i+1} Ay_t||_2^2 + ||M_{i+1} A^T x_t||_2^2$$
$$= \left|\left| A^j A^T (AA^T)^k Ay_t \right|\right|_2^2 + ||(A^T)^j A(A^T A)^k A^T x_t||_2^2$$
$$\leq \lambda_\infty^2 \left( \left|\left| (A^T)^j (AA^T)^k Ay_t \right|\right|_2^2 + ||A^j (A^T A)^k A^T x_t||_2^2 \right)$$
$$\leq \lambda_\infty^2 \left( ||N_i Ay_t||_2^2 + ||M_i A^T x_t||_2^2 \right)$$
$$\leq \Delta_t^i,$$

where for the last inequality we used that $\lambda_\infty \leq 1$. $\qquad\square$

*Proof of Lemma 6:* Given our initialization, $x_0 \in \mathcal{R}(A)$. This implies $x_t \in \mathcal{R}(A), \forall\, t$, due to the update step of the dynamics. Recalling key properties of the matrix pseudoinverse, this implies: $x_t \equiv AA^+ x_t = AA^T (AA^T)^+ x_t$, for all $t$. Similarly, given our initialization, $y_t \in \mathcal{R}(A^T)$, for all $t$, which implies $y_t \equiv A^T (A^T)^+ y_t = A^T A(A^T A)^+ y_t$, for all $t$. Letting $Q = (AA^T)^+$ and $P = (A^T A)^+$, we get the following (where $j = i \mod 2$ and $k = (i - j)/2$):

$$\Delta_t^i = ||M_i A^T x_t||_2^2 + ||N_i Ay_t||_2^2$$
$$= ||M_i A^T AA^T Qx_t||_2^2 + ||N_i AA^T APy_t||_2^2$$
$$= ||M_{i+2} A^T Qx_t||_2^2 + ||N_{i+2} APy_t||_2^2$$
$$= ||A^j (A^T A)^{k+1} A^T (AA^T)^+ x_t||_2^2 + ||(A^T)^j (AA^T)^{k+1} A(A^T A)^+ y_t||_2^2$$
$$= ||A^j (A^T A)^+ A^T (AA^T)^{k+1} x_t||_2^2 + ||(A^T)^j (AA^T)^+ A(A^T A)^{k+1} y_t||_2^2$$
$$= \begin{cases} ||(A^T A)^+ M_{i+2} A^T x_t||_2^2 + ||(AA^T)^+ N_{i+2} Ay_t||_2^2 & \text{if } j = 0 \\ ||(AA^T)^+ M_{i+2} A^T x_t||_2^2 + ||(A^T A)^+ N_{i+2} Ay_t||_2^2 & \text{if } j = 1 \end{cases}$$
$$\leq \max (||Q||, ||P||)^2 \cdot \Delta_t^{i+2},$$

where for the fourth and fifth equality we used the following key property of pseudo-inverses: $A^+ = (A^T A)^+ A^T = A^T (AA^T)^+$. $\qquad\square$

## E   DNA-GENERATION WGAN ARCHITECTURE

| Operation | Kernel | Output Shape | BatchNorm? | Nonlinearity |
|---|---|---|---|---|
| Length of DNA sequence: $L = 6$ | | | | |
| Gradient penalty: $\lambda = 1e^{-4}$ | | | | |
| Batch size: $512$ | | | | |
| $G(z):$ | | | | |
| $z$ | - | 50 | - | - |
| Fully connected | - | 128 | no | tanh |
| Fully connected | - | $16 \times \frac{L}{2}$ | yes | tanh |
| Reshape | - | $16 \times 1 \times \frac{L}{2}$ | - | - |
| Upsampling by 2 | - | $16 \times 1 \times L$ | - | - |
| Convolution | $[1 \times 3] \times 4$ | $4 \times 1 \times L$ | no | tanh |
| $D(x):$ | | | | |
| $x$ | - | $4 \times 1 \times L$ | - | - |
| Convolution | $[1 \times 3] \times 16$ | $16 \times 1 \times L$ | no | tanh |
| Fully connected | - | 32 | no | tanh |
| Fully connected | - | 1 | no | linear |

## F   CIFAR10 WGAN ARCHITECTURE

| Operation | Kernel | Output Shape | BatchNorm? | Nonlinearity |
|---|---|---|---|---|
| Gradient penalty: $\lambda = 10$ | | | | |
| Batch size: $64$ | | | | |
| $G(z):$ | | | | |
| $z$ | - | 100 | - | - |
| Fully connected | - | 1024 | no | LeakyReLU |
| Fully connected | - | 8192 | yes | LeakyReLU |
| Reshape | - | $128 \times 8 \times 8$ | - | - |
| TransposedConv | $[5 \times 5] \times 128$ | $128 \times 16 \times 16$ | yes | LeakyReLU |
| Convolution | $[5 \times 5] \times 64$ | $64 \times 16 \times 16$ | yes | LeakyReLU |
| TransposedConv | $[5 \times 5] \times 64$ | $64 \times 32 \times 32$ | yes | LeakyReLU |
| Convolution | $[5 \times 5] \times 3$ | $3 \times 32 \times 32$ | no | tanh |
| $D(x):$ | | | | |
| $x$ | - | $3 \times 32 \times 32$ | - | - |
| Convolution | $[5 \times 5] \times 64$ | $64 \times 32 \times 32$ | no | LeakyReLU |
| Convolution | $[5 \times 5] \times 128$ | $128 \times 14 \times 14$ | no | LeakyReLU |
| Convolution | $[5 \times 5] \times 128$ | $128 \times 7 \times 7$ | no | LeakyReLU |
| Fully connected | - | 1024 | no | LeakyReLU |
| Fully connected | - | 1 | no | linear |

# G   CIFAR10 GENERATOR IMAGE SAMPLES

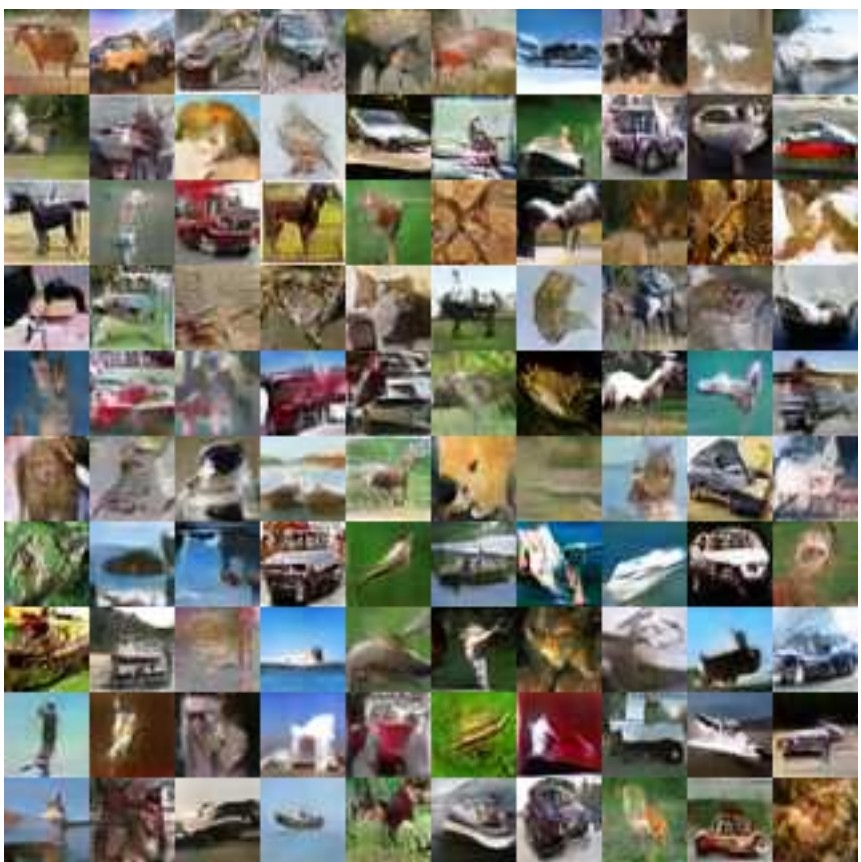

(a) Sample of images from Generator of Epoch 94, which had the highest inception score.

Figure 10: Samples of images from Generator trained via Optimistic Adam on CIFAR10.

## G.1   COMPARISON OF EARLY EPOCH IMAGES OF OPTIMISTIC ADAM VS ADAM

Below we give samples of images from an early epoch 19 Generator trained via Optimistic Adam with 1:1 training ratio, Adam with 1:1 and Adam with 5:1 ratio on CIFAR10. We see that Optimistic Adam has already achieved visually appealing results unlike the latter two vanilla Adam based versions.

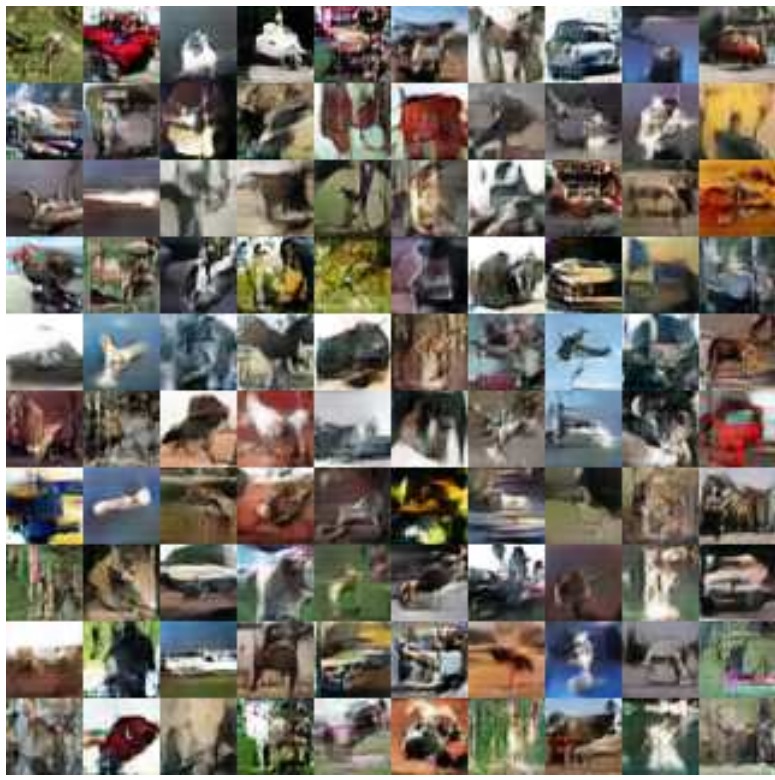

Figure 11: Sample of images from Generator of Epoch 19 trained via Optimistic Adam and 1:1 training ratio.

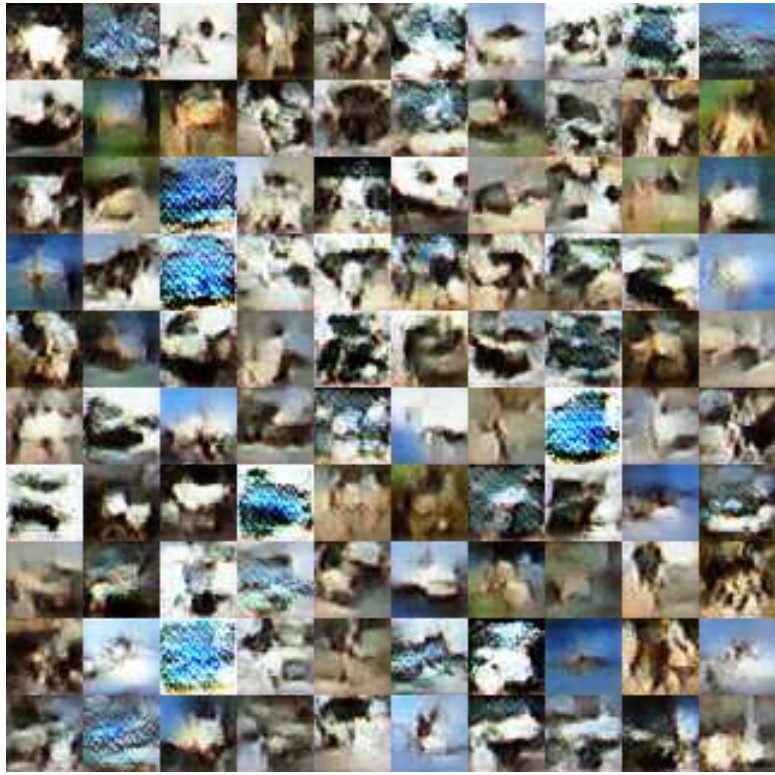

Figure 12: Sample of images from Generator of Epoch 19 trained via Adam and 1:1 training ratio.

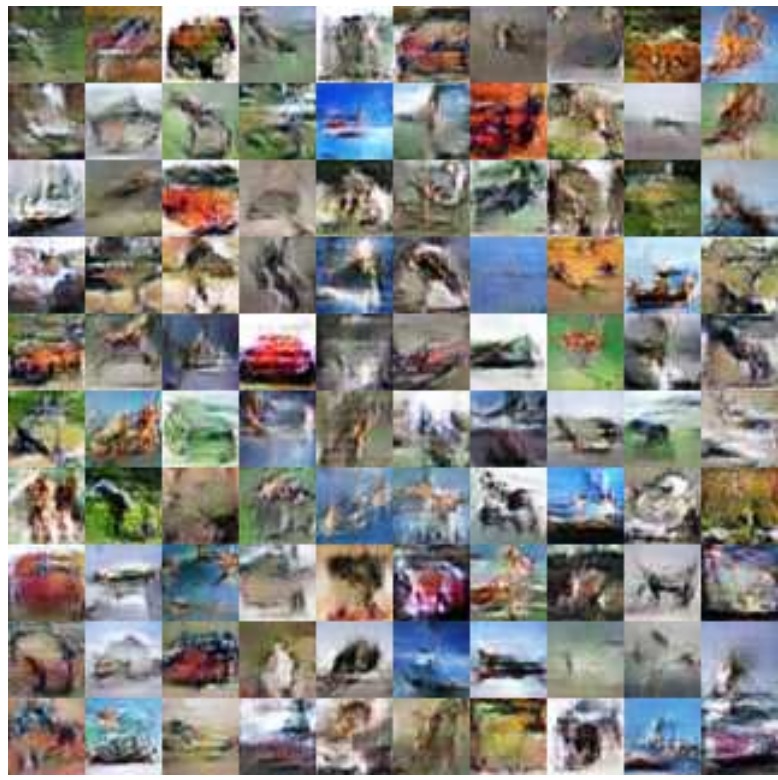

Figure 13: Sample of images from Generator of Epoch 19 trained via Adam and 5:1 training ratio.

# H CIFAR10 ADAM VS. OPTIMISTIC ADAM COMPARISON

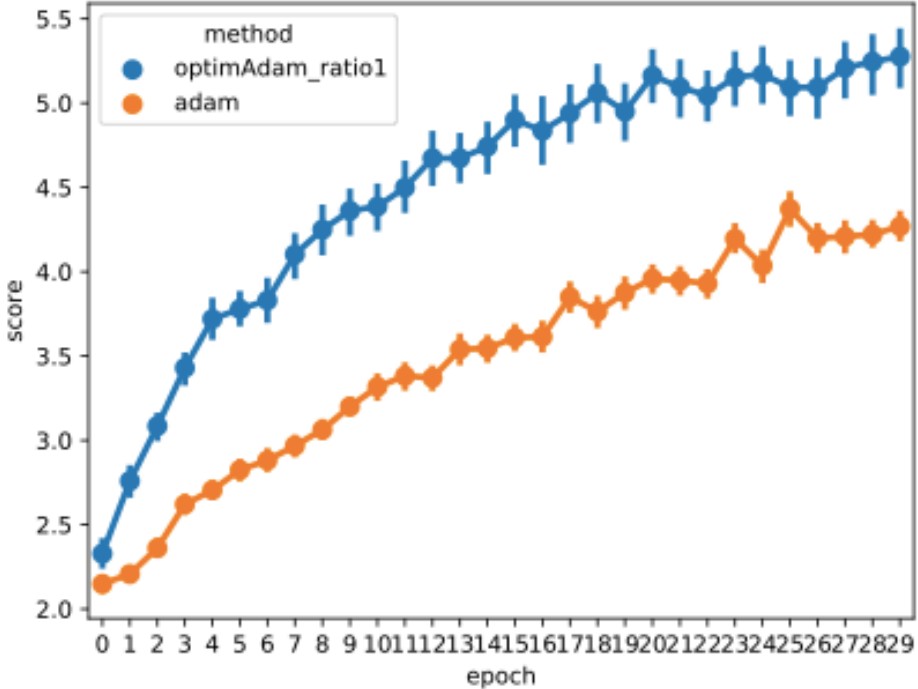

Figure 14: The inception scores across epochs for GANs trained with Optimistic Adam (ratio 1) and Adam (ratio 5) on CIFAR10 (the two top-performing optimizers found in Section 6, with 10%-90% confidence intervals. The GANs were trained for 30 epochs and results gathered across 35 runs.

