# OpenReview forum: "Training GANs with Optimism"
_ICLR.cc/2018/Conference — Accept (Poster)_

### Official Review · AnonReviewer3 · 2017-11-27
**Natural and interesting work with some questionable experimental results**

**Rating:** 7
**Confidence:** 4

**Review:**

This paper proposes the use of optimistic mirror descent to train Wasserstein Generative Adversarial Networks (WGANS). The authors remark that the current training of GANs, which amounts to solving a zero-sum game between a generator and discriminator, is often unstable, and they argue that one source of instability is due to limit cycles, which can occur for FTRL-based algorithms even in convex-concave zero-sum games. Motivated by recent results that use Optimistic Mirror Descent  (OMD) to achieve faster convergence rates (than standard gradient descent) in convex-concave zero-sum games and normal form games, they suggest using these techniques for WGAN training as well. The authors prove that, using OMD, the last iterate converges to an equilibrium and use this as motivation that OMD methods should be more stable for WGAN training. They then compare OMD against GD on both toy simulations and a DNA sequence task before finally introducing an adaptive generalization of OMD, Optimistic Adam, that they test on CIFAR10.

This paper is relatively well-written and clear, and the authors do a good job of introducing the problem of GAN training instability as well as the OMD algorithm, in particular highlighting its differences with standard gradient descent as well as discussing existing work that has applied it to zero-sum games. Given the recent work on OMD for zero-sum and normal form games, it is natural to study its effectiveness in training GANs.The issue of last iterate versus average iterate for non convex-concave problems is also presented well.

The theoretical result on last-iterate convergence of OMD for bilinear games is interesting, but somewhat wanting as it does not provide an explicit convergence rate as in Rakhlin and Sridharan, 2013. Moreover, the result is only at best a motivation for using OMD in WGAN training since the WGAN optimization problem is not a bilinear game.

The experimental results seem to indicate that OMD is at least roughly competitive with GD-based methods, although they seem less compelling than the prior discussion in the paper would suggest. In particular, they are matched by SGD with momentum when evaluated by last epoch performance (albeit while being less sensitive to learning rates). OMD does seem to outperform SGD-based methods when using the lowest discriminator loss, but there doesn't seem to be even an attempt at explaining this in the paper.

I found it a bit odd that Adam was not used as a point of comparison in Section 5, that optimistic Adam was only introduced and tested for CIFAR but not for the DNA sequence problem, and that the discriminator was trained for 5 iterations in Section 5 but only once in Section 6, despite the fact that the reasoning provided in Section 6 seems like it would have also applied for Section 5. This gives the impression that the experimental results might have been at least slightly "gamed".

For the reasons above, I give the paper high marks on clarity, and slightly above average marks on originality, significance, and quality.

Specific comments:
Page 1, "no-regret dynamics in zero-sum games can very often lead to limit cycles": I don't think limit cycles are actually ever formally defined in the entire paper.
Page 3, "standard results in game theory and no-regret learning": These results should be either proven or cited.
Page 3: Don't the parameter spaces need to be bounded for these convergence results to hold?
Page 4, "it is well known that GD is equivalent to the Follow-the-Regularized-Leader algorithm": For completeness, this should probably either be (quickly) proven or a reference should be provided.
Page 5, "the unique equilibrium of the above game is...for the discriminator to choose w=0": Why is w=0 necessary here?
Page 6, "We remark that the set of equilibrium solutions of this minimax problem are pairs (x,y) such that x is in the null space of A^T and y is in the null space of A": Why is this true? This should either be proven or cited.
Page 6, Initialization and Theorem 1: It would be good to discuss the necessity of this particular choice of initialization for the theoretical result. In the Initialization section, it appears simply to be out of convenience.
Page 6, Theorem 1: It should be explicitly stated that this result doesn't provide a convergence rate, in contrast to the existing OMD results cited in the paper.
Page 7, "we considered momentum, Nesterov momentum and AdaGrad": Why isn't Adam used in this section if it is used in  later experiments?
Page 7-8, "When evaluated by....the lowest discriminator loss on the validation set, WGAN trained with Stochastic OMD (SOMD) achieved significantly lower KL divergence than the competing SGD variants.": Can you explain why SOMD outperforms the other methods when using the lowest discriminator loss on the validation set? None of the theoretical arguments presented earlier in the paper seem to even hint at this. The only result that one might expect from the earlier discussion and results is that SOMD would outperform the other methods when evaluating by the last epoch. However, this doesn't even really hold, since there exist learning rates in which SGD with momentum matches the performance of SOMD.
Page 8, "Evaluated by the last epoch, SOMD is much less sensitive to the choice of learning rate than the SGD variants": Learning rate sensitivity doesn't seem to be touched upon in the earlier discussion. Can these results be explained by theory?
Page 8, "we see that optimistic Adam achieves high numbers of inception scores after very few epochs of training": These results don't mean much without error bars.
Page 8, "we only trained the discriminator once after one iteration of generator training. The latter is inline with the intuition behind the use of optimism....": Why didn't this logic apply to the previous section on DNA sequences, where the discriminator was trained multiple times?


After reading the response of the authors (in particular their clarification of some technical results and the extra experiments they carried out during the rebuttal period), I have decided to upgrade my rating of the paper from a 6 to a 7. Just as a note, Figure 3b is now very difficult to read.

---

> ### Author Response · Authors · 2018-01-05
> **Thank you for your comments and how we addressed them in the revision**
>
> We would like to thank you for your comments and suggestions and we explain below how we have addressed your concerns/questions in the updated revision of the paper.
>
> 1) We have replaced limit cycles in the intro with "limit oscillatory behavior". We hope that this term is self explanatory, as we want to avoid further notation for formally defining a limit cycle.
> 2) We have added references for the average converging to equilibrium result (Freund-Ssapire 1999)
> 3) Indeed regret rates, as is typically, always require some boundedness of the optimization space. We have added a sentence on page 3 ("theta and w lie in some bounded convex space") to address this comment.
> 4) We have added a reference on equivalence between GD and FTRL
> 5) Since we are arguing about an equilibrium, it has to be that theta=v is a best response to w. If w > 0, then the best response for theta is minus infinity. If w < 0, then the best response for theta is infinity. If w=0, then any value for theta is a best response. Similarly, at an equilibrium, w also needs to be a best response to theta. If theta>v, then w=infinity is a best response and if theta < v then w=-infinity is a best response. None of the above can be a simultaneous best-response, and the only unique equilibrium is theta=v and w=0.
> 6) We will add a quick sentence about this fact, which follows along the exact same lines as the example above: if y is not in the null space of A, then Ay has some non-zero coordinates. Then the best response for x is to set minus infinity on the positive coordinates of Ay and infinity on the negative coordinates. This will lead to a value of minus infinity, which can be avoided by the y player by choosing a y that lives in the null space of A, leading to a value of zero. Hence, at any equilibrium y is such that Ay = \vec{0} (i.e. the null space of A). Similarly, we can argue for x having to lie in the null space of A^T.
> 7) We have added a convergence rate for Theorem 1 as a function of eta. This result does show that the rate at which Delta_t and consequently the convergence of the solutions goes to the limit value. In particular, this convergence to the limit value of eta gamma^2 Delta_0, happens at an exponential rate of approximately exp{- eta^2 * t / gamma^2}, while this limit value depends linearly with eta. For the regret rates mentioned in section 2 typical values of eta are of the order of 1/T^{1/4} (see e.g. Syrgkanis et al. 2015). Hence, if one wants both regret rates and convergence to equilibrium, these are reasonable values of eta.
> 8) We have added experiments of Adam and optimistic adam in this section too. We thought that Adam was a method particularly useful for image tasks and hence wanted to compare with simpler and more classical algorithms in this section. However, we do admit that we should have also compared with Adam in this section too and we augmented our experiments to include Adam. Indeed adam and optimistic adam performs better in this task too and not only in the image task. Still optimistic adam outperforms adam in this task too.
> 9) Indeed the theoretical results do not imply that an out-of-sample early stopping would work better under optimism than under other methods. However, we wanted to test performance of OMD with an early stopping criterion, since typically, such criteria are used. We indeed are not explicitly doing an early stopping but rather using out-of-sample performance to choose the best iteration. We found this approach to be interesting in practice and grounded in the observations made in Arjovsky et al (as we note in the text before the figure) and we also found that since SOMD was also better performing than other methods other than Adam an interesting finding. Also in terms of last epoch, our updated results show that only Adam has comparable performance with the best performance of optimistic adam or OMD (i.e. with the best learning rates). Also for most learning rates, momentum and nesterov momentum have statistically significant lower performance (indeed comparable, but strictly worse).
> [Continued in the following comment due to character limit]

---

> > ### Author Response · Authors · 2018-01-05
> > **[Continued from last comment due to char limit]**
> >
> > 10) As we explain in the response to reviewer 1, unfortunately this stability of performance wrt to learning rate was only an artifact of a mistake in our implementation and we have removed this comment from the paper.
> > 11) We reran the experiment across 35 runs for 30 epochs (due to compute restrictions) for the two top-performing methods (optimAdam-ratio1 and Adam), and plot the results with 10-90 error bars in the Appendix of the paper, demonstrating that optimAdam indeed reliably performs better in terms of inception score. Once we can run the experiment 100 times for all 100 epochs, we plan to replace our main-text figure with this style of plot.
> > 12) It should indeed apply. We have included such results too. We thought of first comparing with existing proposals and hyperparameter settings in the literature to see the effect of simply adding optimism. However, we agree that we should have included this alternative 1:1 training in this experimental section too. The results in that section are inline again with this intuition and the ratio1 algorithms perform better with their corresponding 5:1 counterparts. (see figure 3b)

---

### Official Review · AnonReviewer1 · 2017-11-28
**Nice paper**

**Rating:** 8
**Confidence:** 4

**Review:**

The paper proposes to use optimistic gradient descent (OGD) for GAN training. Optimistic mirror descent is know to yield fast convergence for finding the optimum of zero-sum convex-concave games (when the players collaborate for fast computation), but earlier results concern the performance of the average iterate. This paper extends this result by showing that the last iterate of OGD also provides a good estimate of the value of bilinear games. Based on this new theoretical result (which is not unexpected but is certainly nice), the authors propose to use stochastic OGD in GAN training. Their experiments show that this new approach avoids the cycling behavior observed with SGD and its variants, and provides promising results in GAN training. (Extensive experiments show the cycling behavior of SGD variants in very simple problems, and some theoretical result is also provided when SGD diverges in solving a simple min-max game).

The paper is clearly written and easy to follow; in fact I quite enjoyed reading it. I have not checked all the details of the proofs, but they seem plausible.
All in all, this is a very nice paper.

Some questions/comments:
- Proposition 1: Could you show a similar example when you can prove the oscillating behavior?
- Theorem 1: It would be interesting to write out the convergence rate of Delta_t, which could be used to optimize eta. Also, my understanding is that you actually avoid computing gamma, hence tuning eta is not straightforward. Alternatively, you could also use an adaptive OGD to automatically tune eta (see, e.g., Joulani et al, "A modular analysis of adaptive (non-)convex optimization: optimism, composite objectives, and variational Bounds," ALT 2017). The non-adaptive selection of eta might be the reason that your method does not outperform adagrad SGD in 5 (b), although it is true that the behavior of your method seems quite stable for different learning rates).
- LHS of the second line of (6) should be theta.
- Below (6): \mathcal{R}(A) is only defined in the appendix.

---

> ### Author Response · Authors · 2018-01-03
> **Thank you for your comments and how we addressed them in the revision**
>
> We would like to thank you for your comments and suggestions and we explain below how we have addressed your concerns/questions in the updated revision of the paper.
>
> 1) In these examples that we give that lead to divergence, it is easy to see that if one takes the step size to zero, then you get a limit cycle (i.e. continuously oscillating behavior). For any other non-zero step-size the behavior is still oscillatory but diverging (i.e. the radius of the cycle is constantly increasing). In some sense, the finite step size, makes the dynamics jump from one limit cycle of the continuous limit dynamics (stepsize=0), to another.
>
> 2) We have added an explicit form of the convergence of Delta_t as a function of eta and gamma in the theorem. Analyzing the limit dynamics with a non-constant stepsize and extending theorem 1 seems feasible, but would complicate even more the inductive proof with extra notation. Hence, we defer such an extension to the full version. It is true that the condition depends on gamma, albeit only an upper bound on gamma is required. If any such upper bound on gamma is known then an appropriate step size can be chosen. Also in terms of an adaptive step size, we believe that our optimistic Adam algorithm is exactly a way of setting and adaptive step size that adapts to the variance of the problem, hence the reason why it out-performs the fixed step size optimistic mirror descent in both experimental sections. So you are right that adaptive step sizes can lead to improved practical performance even in the presence of optimism. Also we note here that in fact there was a small mistake in our implementation of OMD in the DNA experiment which lead to the stability of the performance of OMD across learning rates. We have fixed this mistake and the stability of the performance across learning rates was only an artifact. Still optimism performs better than most methods and optimistic adam leads to the bet last iterate loss, while OMD leads to best early stopping loss.

---

### Official Review · AnonReviewer4 · 2017-12-11
**-**

**Rating:** 6
**Confidence:** 4

**Review:**

This paper proposes a simple modification of standard gradient descent -- called “Optimistic Mirror Descent” -- which is claimed to improve the convergence of GANs and other minimax optimization problems.  It includes experiments in toy settings which build intuition for the proposed algorithm, as well as in a practical GAN setting demonstrating the potential real-world benefits of the method.


Pros

Section 3 directly compares the learning dynamics of GD vs. OMD for a WGAN in a simple toy setting, showing that the default GD algorithm oscillates around the optimum in the limit while OMD’s converges to the optimum.

Section 4 demonstrates the convergence of OMD for a linear minimax optimization problem. (I did not thoroughly verify the proof’s correctness.)

Section 6 proposes an OMD-like modification of Adam which achieves better results than standard Adam in a practical GAN setting (WGANs trained on CIFAR10) .


Cons/Suggestions

The paper could use a good deal of proofreading/revision for clarity and correctness. A couple examples from section 2:
- “If the discriminator is very powerful and learns to accurately classify all samples, then the problem of the generator amounts to solving the Jensen-Shannon divergence between the true distribution and the generators distribution.” -> It would be clearer to say “minimizing” (rather than “solving”) the JS divergence. (“Solving” sounds more like what the discriminator does.)
- “Wasserstein GANs (WGANs) Arjovsky et al. (2017), where the discriminator rather than being treated as a classifier is instead trying to simulate the Wasserstein−1 or earth-mover metric” -> Instead of “simulate”, “estimate” or “approximate” would be better word choices.  And although the standard GAN discriminator is a binary classifier, when optimized to convergence, it’s also estimating a divergence -- the JS divergence (or a shifted and scaled version of it).  Even though the previous paragraph mentions this, it feels a bit misleading to characterize WGANs as doing something fundamentally different.

Sec 2.1: There are several non-trivial but uncited mathematical claims hidden behind “well-known” or similar descriptors. These results could indeed be well-known in certain circles, but I’m not familiar with them, and I suspect most readers won’t be either. Please add citations. A few examples:
- “If the loss function L(θ, w) ..., then standard results in game theory and no-regret learning imply that…”
- “In particular, it is well known that GD is equivalent to the Follow-the-Regularized-Leader algorithm with an L2 regularizer...”
- “It is known that if the learner knew in advance the gradient at the next iteration...”

Section 4: vectors “b” and “c” are included in the objective written in (14), but are later dropped without explanation.  (The constant “d” is also dropped but clearly has no effect on the optimization.)


Overall, the paper could use revision but the proposed approach is simple and seems to be theoretically well-motivated with solid analysis and benefits demonstrated in real-world settings.

---

> ### Author Response · Authors · 2018-01-03
> **Thank you for your comments and how we addressed them in the revision**
>
> We would like to thank you for your comments and suggestions and we explain below how we have addressed your concerns/questions in the updated revision of the paper.
> 1) We changed "solving" to "minimizing"
> 2) We changed "simulate" to "approximate" and also added a small comment to stress that traditional GANs are also a form of metric between two distribution
> 3) We have added a reference for the averages of both players converging to an equilibrium in zero-sum games, in particular Freund-Shapire1999
> 4) We have added the Shalev-Swartz survey on online learning and online convex optimization for the claim that gradient descent is equivalent to FTRL with l_2 regularizer
> 5) We have added the follow-the-perturbed-leader paper of Kalai-Vempala and the lecture notes of Philippe Rigollet for the claim that by knowing the gradient in the next iteration you get constant regret (a consequence of the be-the-leader lemma in these references)
> 6) For vectors b,c,d as we state in the first paragraph of the section we work in the main body only with the simpler game x^TAy and we point that in Appendix D the analysis easily extends to the more complex games with the b, c and d vectors, hence we omitted these vectors in the theorem presented in the main paper. Indeed d is irrelevant for the optimization of both players.

---

### Public Comment · ~Leon_Boellmann1 · 2017-11-08
**OMD on nonconvex optimization**

 Dear authors,
 I have a question on the convergence of OMD. It claims that OMD has a faster convergence rate to the equilibrium of a zero-sum game. Does it hold for an objective function that is convex-concave, or any general objective function? Section 4 only shows the convergence results for a bilinear function. If similar convergence result does not hold for a general objective function,  how does OMD help in the minimax game of GAN, which is generally not convex-concave in the generator and discriminator network parameters?

---

> ### Author Response · Authors · 2017-11-12
> **RE: OMD on nonconvex optimization**
>
> Our main theoretical result in Section 4 is that OMD exhibits last-iterate, rather than average-iterate, convergence in zero-sum games. In particular, its dynamics converges to equilibrium rather than cycling, as gradient descent does, around the equilibrium. We believe that this theoretical guarantee extends to general convex-concave settings.
>
> Inspired by the better behavior of OMD in theory, we evaluate experimentally its performance outside of the convex-concave setting. We observe that it performs better than other methods (e.g. adagrad, adam, nesterov momentum) in adversarial training applications such as training on cifar10 and DNA sequence data.
>
> On the theory front the non convex-concave setting is not well-understood yet. We believe that the theoretical part our analysis could generalize to show convergence to local minimax solutions, but defer this as an interesting open question for future work. We should note that even for non-adversarial training, the non-convex case is not well-understood. Here too methods only have good theoretical properties in the convex setting while still being used in the non-convex setting. One way to view our results it that they complement these results. We show that OMD has last-iterate convergence in the convex-concave setting, and propose using it in the non convex-concave setting where our experimental evaluation shows promising results.

---

> > ### Public Comment · ~Leon_Boellmann1 · 2017-11-12
> > **Thanks a lot!**
> >
> > Thanks a lot for your reply!

---

### Public Comment · (anonymous) · 2017-12-06
**Proof of Lemma 4 in Appendix**

Dear authors,

Could you please provide more details on why the inequalities on the top of page 23 hold true where you try to bound $||x_{t-2}||^2$ and $\Delta^i_{t-2}$? Did the proof of Lemma 4 implicitly assume the induction hypothesis? Because it looks very weird to me. Thanks!

Also, in the proof of the Theorem 3, if we assume all the lemmas are correct, then wouldn't the step from ineq. (40) to ineq. (34) require $\eta > \gamma$ instead of $\eta < \gamma$? If so, then the condition in the main Theorem also should be changed? Correct me if I'm wrong.

BTW, in Theorem 1, matrix A and A^T have the same spectral norm so I guess you can drop the op $\max{A, A^T}$ directly.

---

> ### Author Response · Authors · 2018-01-03
> **Addressing comment of Lemma 4 and updated version of Lemma**
>
> Thank you for your comment, we have uploaded a revision of the paper that we believe will address these concerns:
>
> 1) In the first version of our paper, Lemma 4 did indeed depend on the induction hypothesis for time t-2. This has been resolved in the latest revision, where we have unpacked this implicit use of the induction hypothesis and re-structured the proof a bit. Lemma 4 is currently proving a weaker statement which is still good enough for the final conclusion of the theorem. We have also cleaned up the proof in general and corrected some typos in other parts.
> 2) You are also correct here. This was a typo and should have been 1/gamma rather than gamma. In the new version of the proof, the constants in the conditions and the rates of convergence has slightly changed and we have updated the correct conditions in the revision, while we have also augmented the convergence rates in the theorem statement to explicitly contain the dependence on gamma, which we omitted in the initial submission as we are treating gamma as a constant.
> 3) Indeed we simplified and removed the max operator, since as you say spectral norm is the same for A and  A^T.

---

### Decision · Program_Chairs · 2018-01-29
**ICLR 2018 Conference Acceptance Decision**

**Decision:**

Accept (Poster)

**Comment:**

The reviewers thought the paper provides an interesting line of research.